# The Sustainability of Regional Innovation in China: Insights from Regional Innovation Values and Their Spatial Distribution

## Yipeng Zhang

School of Economics and Resources Management, Beijing Normal University, Beijing 100875, China; 201731410002@mail.bnu.edu.cn

**Abstract:** As the continuous improvement of the quality of innovation becomes increasingly significant for balanced regional development in China, it is critical to provide insights into the sustainability of regional innovation in China from the viewpoint of value. This study estimates regional innovation values based on an improved regional innovation value model incorporating patent values and a regional innovation indicator system. Data for invention patents as well as regional innovation indicators in 282 cities from 1987 to 2019 in China are utilized for estimation. Based on the estimated parameters and Monte Carlo simulation, city-level innovation values are calculated as benchmarks, along with provincial and regional innovation values, to analyze the patterns of the spatial distribution and agglomeration of regional innovation value. The findings are as follows. (1) The regional innovation value model provides an effective way to measure regional innovation in terms of value. (2) The regional innovation values are unevenly distributed; cities with higher innovation values are clustered in Eastern China, while most other cities have much lower innovation values. (3) The innovation values in Eastern China are notably higher, and the differences in innovation values between Eastern China and other regions are large and show a trend of first widening and then narrowing during the sample period. (4) The sustainability of regional innovation is not widely achieved, since highly concentrated innovation value is found in only a few regions in the eastern coastal areas. These findings suggest that promoting China's innovation capacity and the sustainable development of technological innovation requires continually implementing innovation-driven development strategies, cultivating high-value innovation, optimizing industrial transfer, improving the layout of the national research infrastructure, giving full play to spatial spillover effects, and promoting interregional innovation information exchange in order to achieve the balanced and sustainable development of regional innovation.

**Keywords:** regional innovation value; reginal innovation indicator; spatial distribution of innovation value; agglomeration of innovation value; sustainability of regional innovation

## 1. Introduction

China has attached great importance to the development of technological innovation and regards the innovation-driven deployment strategy as a core national strategy. For instance, the report from China's 20th National Congress highlighted the importance of innovation in the country's modernization and the need to accelerate scientific and technological self-reliance and improvement. The implementation of a series of important policies regarding technological and scientific innovation has enhanced China's innovation capability in recent decades. In recent years, China's ranking in the Global Innovation Index (GII) kept increasing and moved up to 11th in 2022 (see the Global Innovation Index 2022, available at https://www.wipo.int/global_innovation_index/en/2022/ (accessed on 30 April 2023)). In addition, the dramatic increase in the amount of Chinese innovative outputs under the influence of a series of policy strategies has drawn scholars' attention [1,2]. As a matter of fact, Chinese domestic invention patent applications increased from 4065

in 1985 to 1.24 million in 2019 (according to the China Intellectual Property Statistical Yearbooks 1985 and 2019, available at https://www.cnipa.gov.cn/col/col61/index.html# mark (accessed on 30 April 2023)), and Chinese applicants have been filing the most Patent Cooperation Treaty (PCT) applications since 2019 (according to the Patent Cooperation Treaty Yearly Review 2020–2023, available at https://www.wipo.int/publications/en/ details.jsp?id=4666&plang=EN (accessed on 30 April 2023)).

However, rapid growth in the amount of innovation outputs does not necessarily ensure the sustainable growth of innovation capacity in China. Although the policies of establishing national high-tech industrial parks and innovation cities, the growth of direct foreign investment, and the amendments to patent laws may have contributed to the surge in innovation outputs in China [3–6], evidence has shown that the majority of regions in China are struggling with low innovation capability and are clustered with other regions of low innovation capability [7], even in the regional high-tech industries [8], which suggests an unsustainable development path of innovation in the majority of regions in China. To explore the sustainability of innovation in more detail, it is essential to analyze innovation from another prospective rather than the number of innovative outputs. Quantitative indicators of innovation based on the assumption of homogeneity and additivity ignore technological differences between innovative outputs and hardly reflect the true technological level [9]. Instead, the quality of innovation is the essence of endogenous economic growth [10]. In addition, the literatures suggest that the value representing the economic return can be an effective indicator of the quality of an innovative output, such as a patent [9,11,12].

Another manifestation of China's unsustainable development of technological innovation is the uneven spatial distribution of its innovation capability. Large regional differences in economic endowment result in significant gaps in innovation capacity between regions, as measured by the number of patents, clearly reflecting uneven regional distribution [13–15]. However, such studies primarily utilize quantitative indicators of innovation, which may underestimate the gaps in regional innovation due to the underlying assumption of technological homogeneity in the innovative outputs. Other studies have drawn similar conclusions when utilizing data for R&D expenditure, human capital, government support, technology conversion expenditure, and other factors as proxies for the innovation in a region [8,16–18]. However, such proxies may not properly represent innovation, as it should be the result of the transformation of these inputs [19], which may lead to bias in analyzing the spatial distribution of regional innovation. Therefore, the research on the spatial distribution of regional innovation measured in terms of value in China remains relatively scarce, necessitating further investigation.

As such, this study aims to answer the following questions. How can the regional innovation in China be accurately measured in terms of the values of innovation outputs? What is the spatial distribution of regional innovation value in China? Using data from Chinese invention patents, this study measures regional innovation value using an improved regional innovation value model. Then, a spatial measurement method is used to explore the spatial characteristics of regional innovation value and analyze the sustainability of regional innovation in China. This study is organized as follows. Section 2 provides a review of the relevant literature and presents the hypotheses for this paper. Section 3 presents the methodology and data utilized to estimate and analyze the value of regional innovation. Section 4 contains the empirical results and discussions of the estimation and spatial analysis of the regional innovation value. Section 5 concludes the findings and discussions about the regional innovation value in China and provides policy recommendations. Section 6 discusses the limitations in this paper and potential interests for future research studies.

This study's main contributions are as follows. (1) It establishes an improved model that incorporates patent value and regional innovation factors to provide a more accurate measure reflecting regional innovation values. (2) Micro-level patent data are combined with macro-level data on regional innovation factors to provide a more comprehensive

estimation of regional innovation values. (3) The spatial econometric analysis of regional innovation values at the city, province, and country levels in China can enrich the research on the spatial distribution of regional innovation, in contrast to analyzing the spatial distribution of innovation based on quantitative indicators.

## 2. Literature Review and Research Hypotheses

### 2.1. The Value of Innovation

The value of innovation refers to the economic value of the innovation output, including the gains from owning a patent right, the profit from promoting new products and services, and the external benefits from scientific and technological innovations. In this paper, the innovation value in a region is defined as the aggregation of the (discounted) net returns from all invention patents in the region. The reasons why the other two sources of innovation value are not addressed are twofold: the data for the sales of new products and services are not generally accessible across regions in China, and the method to appropriate the social value of innovation is limited [20]. Although not all technological innovations are patentable, and many patented inventions might not be innovative enough [21,22], the enriched patent information provided by patent databases and the availability of high-quality computers and software today have encouraged researchers to use patent data to study innovation [19].

As a main innovation output, patents contain a wealth of information about new technologies and have, therefore, received a great deal of research attention. However, merely using the number of patents to measure innovation might be deceptive because the technological innovativeness endowed by a patent and its economic effect might not be significant [22], which may result in measurement noise [9]. Therefore, some scholars have taken a different approach by estimating the value of patents based on the behavioral characteristics of patent rights holders and the validity period of patents. A primary advantage of the patent value is that it provides a monetary measure of an innovative output's exclusive market profit enforced by the patent system [11,23], which allows a more objective and precise measure of innovation and comparative analyses between countries' innovation values [9,24]. However, such methods often neglect the comprehensive effect of other factors influencing the innovation processes in a region and may not address the spatial inequality of innovation between regions, which raises the need for a measure of innovation that embodies the advantages of patent values while balancing the comprehensive effects of factors influencing regional innovation. A potential solution to this issue can be found in studies that construct indicator systems to measure innovation and analyze its spatial distribution [8,25–27]. Although the primary focus in these studies is not the value of regional innovation, they indicate that an indicator system addressing the input factors, innovation environmental factors, and other economic factors can effectively integrate the impacts of these factors on regional innovation. Therefore, this paper aims to establish a regional innovation value model that incorporates the patent value estimation and a regional innovation indicator system consisting of the input, innovation environmental, and economic factors to effectively estimate the regional innovation value in China. As such, this paper proposes the first research hypothesis:

**Hypothesis 1 (H1).** *The regional innovation value model that incorporates the patent value and a regional innovation indicator system can effectively estimate the regional innovation values in China.*

### 2.2. The Spatial Distribution of Regional Innovation

In recent years, the study of the spatial characteristics and correlations of innovation across geographical regions has drawn increasing attention. The relevant studies may have originated from the theoretical framework of national innovation system (NIS) that was established by scholars including Freeman [28], Lundvall [29], and Nelson [30]. In the 1990s, Cooke and his colleagues [31,32] introduced the notion of the regional innovation system

(RIS), which synthesized the framework of the NIS and empirical evidence of regional innovation. Accordingly, the theory of the RIS proposed that economic agents' activities directly rely on the characteristics of the region, and through the integration of financial capacity, institutionalized learning, and productive culture, the economic agents conduct systemic innovative activities [32]. Therefore, the outcomes of innovative activities in a region are not only the results of the agents' own effort but also under the influence of regional innovation and economic environmental factors.

Based on the RIS theory, scholars have conducted a series of empirical studies examining the characteristics of the spatial distribution and correlation of regional innovation in China. For instance, Li [13] established a set of stochastic frontier models and used quantitative indicators of patents to examine the disparity in innovation performance between Chinese regions, and the author confirmed that while the innovation environment exerted significant impacts on innovation efficiency, the overall innovation efficiency between regions became increasingly disparate. Xu et al. [7] utilized Chinese patent information at the province level to conduct research on the inequality of regional innovation in China based on concentration indexes and concluded a high degree of inequality of regional innovation accompanied by the regional agglomeration effects. Tu et al. [8] incorporated Moran's index and LISA cluster maps with a comprehensive indicator system for the innovation capacity of regional high-tech industries and found that the spatial distribution of regional high-tech industries' innovation capacity was vastly unbalanced and characterized by low–low agglomeration. Although all of these studies provided evidence on the unbalanced spatial distribution of regional innovation in China, they mainly utilized the quantitative indicator (or combined with an innovation indicator system) as the measure of regional innovation, which may cause the previously mentioned challenge. Moreover, the small number of other studies that applied the method of the patent value to analyze the value of innovation in China were limited to the country level [23,33], province level [34], or industrial level [12,23,35], and few studies have addressed the patent value at the city level and conducted an analysis of the spatial distribution of innovation values by integrating the city, province, and country levels. As the quality of innovation becomes increasingly significant for balanced regional development in China, it is critical to provide insights into the spatial distribution of regional innovation in China from the viewpoint of value. In addition, since the value of innovation can address the heterogeneity of technology embodied in innovative outcomes [9,11,12], there may be an even larger regional disparity of innovation values in China. Therefore, this paper puts forward the following hypothesis:

**Hypothesis 2 (H2).** *The spatial distribution of regional innovation value is unbalanced and mainly characterized by the agglomeration of regions with low innovation values.*

## 3. Methods and Data

### 3.1. Methodology

In this paper, an improved regional innovation value model was established based on the original patent renewal model and a regional innovation indicator system. The regional innovation indicator system was identified, and the weighted valued for the main (tier 1) indicators in 282 sample cities in China from 1987 to 2019 were estimated using the entropy weight method. Moreover, the nonlinear least squares method was employed to estimate the essential parameters in the regional innovation value model that incorporated both datasets for approximately 2.8 million invention patents applied for and granted in China and the estimated weighted indicators of regional innovation. The fitted parameters based on the empirical results were utilized in a series of Monte Carlo simulation processes to estimate the innovation values at the city, province, and area levels in China. The analyses of the spatial distribution of regional innovation values in China were then conducted using a local Moran's index, LISA cluster maps, and hotspot maps in order to examine the sustainability of regional innovation in China.

Original Model: Patent Renewal Model

(1) Setup of the patent renewal model

Researchers have often used patent renewal information to estimate the patent value (some other information, such as the patent family size (e.g., [36,37]) and citations (e.g., [37,38]), can also be used as indicators of patent quality). The underlying logic of this method is that under the patent system, the holders of patent rights must periodically pay renewal fees to maintain the right to receive exclusive economic benefits from the patent. Thus, it is assumed that the number of periods a patent right remains effective is positively related to its economic value. Pakes and Schankerman originated the patent renewal model that formalizes this relationship [39]. This model was further developed by Schankerman and Pakes [9], Lanjouw [36], Pakes [40], and Bessen [11]. An advantage of the patent renewal model is that it establishes the theoretical mapping of the patent value onto the observable renewal behavior of patent holders, which then allows for the estimation of the patent value. Moreover, the aggregated patent value within a region (or nationwide) is often considered an estimate of the regional or national innovation value.

The paten renewal model assumes that a representative agent who holds a patent of cohort $i$ determines whether to renew the patent at age $t$ (years renewed) in order to maximize its discounted economic value (i.e., the discounted net return) throughout the renewal period. A simplified version of this discrete optimization problem can be expressed as the following:

$$\max_{T\in\{1,2,\dots,\overline{T}\}} \sum_{t=1}^{T} \lambda^t (R_{i0} \cdot d^{-t} - C_{it}), \tag{1}$$

where $R_{i0}$ represents the initial implicit return to the patent right and $d$ represents the exogeneous decay rate (equivalently, studies also use the depreciation rate $\delta_{it}$ to express the decay of a patent's return (i.e., $d_{it} = 1 - \delta_{it}$)) of the patent's return. Thus, the term $R_{i0} \cdot d^{-t}$ represents the nonincreasing implicit return to the patent rights at age $t$ of the patent. Since the return to the patent rights cannot be observed directly, scholars have establised different setups modelizing the dynamics of the return. One is to assume that the return changes deterministically with an initial $R_{i0}$ and a series of exogenous decay rates (e.g., [12,33,34]), which can be further simplified by assuming the indifference between decay rates, as in this paper. The other is to assume a stochastic dynamic of the return (e.g., [23,40,41]). Bessen, however, claimed that the stochastic dynamics of the return do not necessarily distinguish the estimation results [11]. This paper, therefore, assumes the deterministic dynamics of the return. Furthermore, $\lambda$ is the discount factor, $T$ is the optimizer the representative agent chooses to maximize the overall discount net return of the patent, and $\overline{T}$ represents the statutory limit on the patent protection (20 years). In China, the statutory limit on patent protection was originally 15 years for invention patents. This was changed to 20 years with the second amendment of the Patent Law of the People's Republic of China on 1 January 1993. Here, $C_{it}$ is the annual renewal fee in the same period (to guarantee the existence of an optimal renewal period $T*$ in a deterministic setup, the flow of annual implicit returns $R_{i0} \cdot d^{-t}$ must be nonincreasing in age $t$, while the flow of annual renewal fees $C_{it}$ must be nondecreasing in $t$ so that the flow of net returns converges to 0 in time), and the schedule of annual renewal fees is often specified by each country's patent office (the schedule of renewal fees for invention patents applied in China and its amendments are published by the China National Intellectual Property Administration and can be found at https://www.cnipa.gov.cn/ (accessed on 31 May 2022)) and is nondecreasing in a patent's renewal period.

Since the returns of the patent are nonincreasing and the costs of renewal are nondecreasing in the renewal period, it is clear that the representative agent would maximize the discount net return at an optimal renewal period $T*$, such that any additional period of renewal would result in a net loss in that period:

$$R_{i0} \cdot d^{-T*} - C_{iT*} > 0, \text{ and } R_{i0} \cdot d^{-(T*+1)} - C_{i,T*+1} \leq 0 \text{ for } T* \in \{1, 2, \dots, 20\}. \tag{2}$$

If there exists no such *T\** within the statutory limit, then the representative agent finds the statutory limit as the optimizer (i.e., there exists a corner solution at the upper boundary of the renewal period).

The solution to this optimization problem indicates that any holder of the patent in cohort *i* would keep renewing the patent rights as long as the current return (taking account of its decay) exceeds the renewal fee. Accordingly, the observable renewal proportion of all patents in cohort *i* at a certain renewal period *t* can be expressed as a function of the initial return $R_{i0}$, along with other exogenous factors (i.e., the corresponding decay rate *d*, renewal period *t*, and renewal fee $C_{it}$). Although the initial return of a patent in a cohort is not observable, it can be represented by a random variable following the distribution characterized by a particular probability density function and parameters associated with the specific cohort. Moreover, studies have shown that the distribution of patent values is highly skewed (e.g., [9,11,42,43]). In the framework of the patent renewal model, the lognormal distribution has been confirmed to have the best fit for renewal data in studies focused on Europe and the US (e.g., [9,11,44]). In studies on Chinese patent value, it is often accepted that the initial returns of the patent value are distributed lognormally (e.g., [12,23,33]). In the following sections of this paper, a lognormal distribution is used as the distribution method of the initial returns of the patents. This study tests three candidates for the distribution of the initial returns of sample invention patents, namely lognormal, Pareto–Levy, and Weibull distributions. The econometric model that best fits the sample data is the one that assumes a Pareto–Levy distribution according to the adjusted $R^2$ and the weighted sum of squared errors, suggested by Schankerman and Pakes [9]. The result is consistent regardless of the method of identifying the cohorts of sample patents. However, the estimated shape parameters (often denoted by α) in different sample sets are somewhere in between 0.373 and 0.436 and fall within the range of [0, 1]. This means that the mean of the patent value is infinity in principle and will lead to overestimating both the patent value and regional innovation value. To avoid overestimation, this study chooses the distribution that yields the second-best fitness, which is the lognormal distribution. Here, $R_{i0} \sim LN(\mu_i, \sigma_i)$ or $\ln R_{i0} \sim N(\mu_i, \sigma_i)$ is used, where $\mu_i$ and $\sigma_i$ denote, respectively, the population mean and standard deviation of the initial value of patents in cohort *i*. In addition, following Bessen [11], Zhang et al. [12], and Og et al. [45], the log of the initial return of a patent is a function of some observable characteristics of the patent. In this study, the selected characteristics include the year and city of the patent application, the type of applicant, and the industrial classification of the patent, which are represented by $x_i$, the vector of dummy variables. It then follows that:

$$ri0 = \ln R_{i0} = \mu_{i0} + \boldsymbol{\beta}' \boldsymbol{x_i} + \zeta_i, \tag{3}$$

where $\mu_{i0} + \boldsymbol{\beta}' \boldsymbol{x_i}$ is the mean of $R_{i0}$'s distribution (i.e., $\mu_{i0} + \boldsymbol{\beta}' \boldsymbol{x_i} = \mu_i$). Here, $\boldsymbol{x_i}$ is the vector of variables controlling the factors that influence the initial returns of the patents in cohort *i*. As suggested by previous studies, factors including the categorical variables identifying the cohort, the inputs for innovative activities, and other innovation and economic environmental factors may exert significant impacts on the initial returns [9,11,12,23,24,33]. Here, $\boldsymbol{\beta}$ is the vector representing the coefficients of these factors and $\zeta_i$ denotes the normally distributed stochastic error with a zero mean and standard deviation $\sigma_i$. To further simplify the analysis, the standard deviation is assumed to be indifferent across cohorts [9–11] (i.e., $\sigma_i = \sigma_j = \sigma$ for all *i* and *j*). Therefore, according to the optimization condition, the proportion of patents in cohort *i* that are renewed at period *t* can be formalized as the following:

$$Pit = 1 - \Pr\left[\frac{ri0 - (\mu i0 + \boldsymbol{\beta}' \boldsymbol{x_i})}{\sigma} > \frac{cit - t \ln d - (\mu i0 + \boldsymbol{\beta}' \boldsymbol{x_i})}{\sigma}\right] = 1 - \Phi\left[\frac{cit - t \ln d - (\mu i0 + \boldsymbol{\beta}' \boldsymbol{x_i})}{\sigma}\right], \tag{4}$$

where $c_{it} = \ln C_{it}$ as the log transformation of the renewal fee for patents in cohort *i* at period *t*. Note that the renewal condition (indicated by the left inequality in (2)) is normalized according to the assumption of a lognormal distribution of the initial return (Equation (3)), and it is clear that $(r_{i0} - (\mu_{i0} + \boldsymbol{\beta}' \boldsymbol{x_i}))/\sigma$ follows the standard normal distribution $N(0,1)$, whose cumulative density function is denoted by $\Phi(\cdot)$.

Based on Equation (4), an econometric model that links the patent holder's renewal behavior to the patent value can be established as the following:

$$y_{it} \equiv \Phi^{-1}(1 - P_{it}) = \frac{1}{\sigma}[-(\mu_i 0 + \boldsymbol{\beta}' \boldsymbol{x_i}) + c_i t - t \ln d] + \varepsilon_{it}, \tag{5}$$

where $y_{it}$ is defined as the inverse of the cumulative density function of the previously described standard normal distribution; $P_{it}$ stands for the proportion of patents in the $i$th cohort that are renewed at period $t$; $\boldsymbol{x_i}$ is the vector of variables controlling the initial return of the patent in the same cohort; $c_{it}$ is the log of the renewal fee; $t$ represents the renewal period; $\mu_{i0}$, $\sigma$, $\boldsymbol{\beta}$, and $d$ are the previously defined parameters defined to be estimated; $\varepsilon_{it}$ is the error term. Using the nonlinear least squares method, one can estimate the parameters that are key to calculating the patent value.

(2) Calculation of the patent value

The estimates based on Equation (5) contain the fitted parameters describing the patents' initial distributions (i.e., $\hat{\mu}_{i0}$, $\hat{\boldsymbol{\beta}}$, and $\hat{\sigma}$) and the decay rate of the patent value with age (i.e., $\hat{d}$). Using the fitted parameters as well as the given schedule of renewal fees for patents in the $i$th cohort, one can use a Monte Carlo simulation to generate a set of pseudo-random numbers representing the initial returns of patents in the $i$th cohorts (i.e., $\widetilde{R}_{i0}$) and simulate their renewal decisions, which results in the optimal renewal periods ($\widetilde{T}^*$) for each pseudo-random number, satisfying the following condition:

$$\widetilde{R}_{i0} > C_{i\widetilde{T}^*} \cdot \hat{d}^{\widetilde{T}^*} \text{ and } \widetilde{R}_{i0} > C_{i,\widetilde{T}^*+1} \cdot \hat{d}^{\widetilde{T}^*+1}, \tag{6}$$

where $\widetilde{T}^* = 1, 2, \ldots, \overline{T}$. Then, the discount value (the discount factor, $\lambda$, is set to 0.95 as suggested by Deng [41]) of each simulated pseudo-number is:

$$\widetilde{V}\left(\widetilde{T}^*\right) = \sum_{t=1}^{\widetilde{T}^*} \lambda^t \left(\widetilde{R}_{i0} \cdot \hat{d}^{-t} - C_{it}\right). \tag{7}$$

The last step is to calculate the arithmetic average of the simulated discount values within each group of simulated optimal renewal periods; that is, to calculate the average of all $\widetilde{V}\left(\widetilde{T}^*\right)$ in Equation (7) for those whose age is $\widetilde{T}^*$. Then, the result is the estimate of the value of the patents in the $i$th cohort at age $t$.

### 3.2. Extended Model: Regional Innovation Value Model

A limitation of the original patent renewal model is that it does not incorporate the comprehensive effects of regional factors that might influence innovation and its value, which might lead to estimation bias when aggregated at the regional level. Innovation activity in a region is a systematic process this is potentially affected by various factors [46]. Not only do innovation inputs determine the outcome of innovation activity but the region's innovation environment (e.g., government fiscal support and educational effort) and economic environment (e.g., market scale and financial development) are also essential for innovation behaviors, as well as the conversion of new ideas and inventions into regional productivity. The comprehensive effects of these factors determine the regional innovation capacity and the differences across regions [47–49]. Although the choices of particular indicators representing the factors of innovation inputs, the innovation environment, and the economic environment may differ in the literature, scholars who focus on the regional innovation in China often incorporate indicators representing innovation inputs, the innovation environment, and the economic environment in their indicator systems and find them statistically significant in analyzing the spatial inequality of regional innovation in China [8,25,27], although the particular choices of indicators may differ in the relevant studies. The average and overall levels of the regional innovation value may, therefore, differ owing to such factors. However, such factors are not directly reflected in the patent renewal model. In addition, although some studies have controlled some of the factors in the patent renewal model (e.g., [9,11,50]), the lack of a systematic measure of regional

innovation factors and their effects on the patent value might result in the insufficient measurement of regional innovation value.

(1) Regional innovation value model

To include the comprehensive effects of regional innovation factors to make suitable estimations of regional innovation value, this study modifies the patent renewal model by introducing a regional innovation indicator system using Jaffe's knowledge production function [51,52]. In this way, an extended regional innovation value model is established.

According to the knowledge production function, the output of innovation activity can be expressed as a Cobb–Douglas production function of certain factors. This study primarily addresses the previously mentioned three sets: the innovation input, the environment for innovation activity, and the region's economic environment. Thus, the log of the initial returns of a patent can be expressed as:

$$r_{i0} = \mu_{i0} + b_1 \ln Inp_{i,L2} + b_2 \ln Env_{i,L2} + b_3 \ln Eco_{i,L2} + v_i, \tag{8}$$

where $Inp_{i,L2}$, $Env_{i,L2}$, and $Eco_{i,L2}$ denote the indicators of the innovation input, innovation environment, and economic environment two periods prior to a patent application by the $i$th cohort in their associated regions. Studies have shown that there are time lags between a firm's R&D investment and its innovation output [53–55]. However, the actual time lags remain debatable (see [56]). In addition, the samples in this study include invention patents not applied for by firms. Therefore, a relatively short time lead (2 years) in the indicators prior to the patent application is chosen for the model. Here, $b_1$, $b_2$, and $b_3$ are the elasticities of the indicators; $\mu_{i0}$ denotes a constant; and $v_i$ denotes an error term following normal distribution with a zero mean and standard deviation $\sigma$.

Equation (8) indicates that the innovation inputs (along with the innovation and economic environmental factors) lead to the lagged outputs of innovation. However, the direction of causality between the innovation inputs and the value of the innovation outputs has not been systematically studied in the existing literature. In the related literature, the direction of the causality is somewhat ambiguous. There exists evidence indicating that the direction of causality is from R&D investment to firm profitability [57], while other studies have found insignificant causal relationships between the inputs and outputs of innovation in particular industries (e.g., [58]). Since the primary focuses in this paper are the estimation of the regional innovation value and the analysis of the spatial distribution of the regional innovation value in China, the theoretical relationship between the value of innovation outputs and inputs remains as presented in Equation (8), leaving the analysis of the causality for potential future studies.

Based on Equation (8), the proportion of patents renewed at age $t$ can be expressed as:

$$P_{it} = 1 - \Phi\big[\frac{c_{it} - t \ln d - (\mu_{i0} + b_1 \ln Inp_{i,L2} + b_2 \ln Env_{i,L2} + b_3 \ln Eco_{i,L2})}{\sigma}\big]. \tag{9}$$

Rewriting the above equation as in Equation (5) yields:

$$y_{it} \equiv \Phi^{-1}(1 - P_{it}) = \frac{1}{\sigma}\big[-(\mu_{i0} + b_1 \ln Inp_{i,L2} + b_2 \ln Env_{i,L2} + b_3 \ln Eco_{i,L2}) + c_{it} - t \ln d\big] + \varepsilon_{it}, \tag{10}$$

which is the basic regional innovation value model.

Furthermore, the influence of the patents' observable characteristics as described in Equation (3) is also considered in the following:

$$y_{it} = \frac{1}{\sigma}\big[-(\mu_{i0} + b_1 \ln Inp_{i,L2} + b_2 \ln Env_{i,L2} + b_3 \ln Eco_{i,L2} + \boldsymbol{\beta}'\boldsymbol{x_i}) + c_{it} - t \ln d\big] + \varepsilon_{it}. \tag{11}$$

Finally, some variation in the decay rate is allowed in the model, considering that technology in general is evolving at an accelerating pace, which might increase the depreciation rate of the innovation. As suggested by Schankerman and Pakes [9], at age $t$, the decay rate of the value of the patents applied for in year $r$ can be expressed in an exponential form:

$$d_{rt} = d \exp(\theta_1 D_{rt}^1 + \theta_2 D_{rt}^2), \tag{12}$$

where $D_{rt}{}^1$ and $D_{rt}{}^2$ are dummy variables representing the actual year (i.e., $r + t$), which satisfy:

$$D_{rt}^1 = \begin{cases} 1 \ if \ 2001 \leq r+t \leq 2010 \\ 0 \ \text{elsewhere,} \end{cases},$$
$$D_{rt}^2 = \begin{cases} 1 \ if \ 2011 \leq r+t \leq 2019 \\ 0 \ \text{elsewhere.} \end{cases}. \tag{13}$$

Therefore, the regional innovation value model can be written as:

$$y_{it} = \frac{1}{\sigma}[-(\mu_{i0} + b_1 \ln Inp_{i,L2} + b_2 \ln Env_{i,L2} + b_3 \ln Eco_{i,L2} + \boldsymbol{\beta'x_i}) + c_{it} - t \ln d]$$
$$- \frac{1}{\sigma}(\theta_1 \sum_{\tau=1}^{t} D_{r\tau}^1 + \theta_2 \sum_{\tau=1}^{t} D_{r\tau}^2) + \varepsilon_{it} \tag{14}$$

Equations (10), (11) and (14) are the three functions of the regional innovation value model that are incorporated into the effects of innovation factors in the associated regions. However, to complete the model, a regional innovation indicator system needs to be specified.

(2) Regional innovation indicator system

In general, the theory of knowledge production regards the innovative outputs as a function of the inputs, including R&D investment, human capital, and knowledge accumulation [51,52]. Li [13] suggested that patents can also be treated as a function similar to the knowledge production function. Among the studies that used indicators systems of regional innovation, the indicators representing the inputs of innovation were often included along with patent information (e.g., [8,59,60]). Besides the innovation inputs, the innovation and economic environment are also essential to the technological innovation in a region. As the theory of RIS indicates, regional innovation is a series of systemic activities under the comprehensive effects of local innovation and economic environmental factors such as governmental effort, financial support, and institutional support. Following the logic of the RIS theory, empirical studies have also aggregated the innovation and economic environmental effects on regional innovation by addressing relevant factors in the indicator systems [60]. This paper extracts indicators that potentially characterize the innovation input, innovation environment, and economic environment, which determine the output of regional innovative activities. A regional innovation indicator system is established based on the indicator systems suggested in the "Evaluation Report on China's Regional Innovation Capacity 2021" [61] and the "Evaluation Report on the Innovation Capacity of National Innovative Cities 2021" [62], as well as other studies on regional innovation (e.g., [8,59,60]). In addition, due to the limited accessibility of the corresponding data in the sample cities and the timespan, some of the indicators included in the abovementioned reports and studies are excluded, while others are substituted by their alternatives, which then leads to the regional indicator system consisting of three main indicators (tier 1 indicators) and 12 subindicators (tier 2 indicators), as presented in Table 1.

**Table 1.** Regional innovation indicator system.

| Tier 1 Indicators | Tier 2 Indicators | Unit of Measure |
|---|---|---|
| Innovation Input | Number of R&D personnel per 10,000 people | Number/10,000 people |
| | Government R&D spending as a proportion of fiscal expenditure | % |
| | Number of papers published in domestic academic journals per 10,000 people | Number/10,000 people |
| | Number of papers published in foreign academic journals per 10,000 people | Number/10,000 people |
| Innovation Environment | Government educational spending as a proportion of fiscal expenditure | % |
| | Ratio of telephone, mobile phone, and Internet subscribers to population | Ratio |
| | Number of national-level incubators for innovative enterprises | Number |
| | Number of patent agencies | Number |
| Economic Environment | GRP per capita | Yuan/person |
| | Average wage of employees | Yuan/person |
| | Amount of foreign capital actually used per capita | Yuan/person |
| | Ratio of total market value of listed enterprises to GRP | % |

First, the regional innovation input factors mainly include human capital, capital, and accumulated knowledge. As suggested by Pan et al. [60], the R&D personnel, expenditure invested, and knowledge accumulated within a region are essential indicators to



measure the abundance of resources to produce innovative outcomes. Studies often use the number of R&D personnel, the amount of internal R&D expenditure, and the number of academic papers published in a region as indictors representing these inputs [8,60–62]. However, the data for the amount of internal R&D expenditure by local enterprises are not accessible in most cities, especially in the early sample years. A potential alternative is the R&D expenditure by the local government, which indicates the local government's direct spending on innovation projects. In addition, to reduce the potential estimation bias caused by endogeneity, the indicator for R&D personnel is calculated per 10,000 people and the government R&D expenditure is converted to its proportion to the local fiscal expenditure, as suggested in the relevant literature (e.g., [63–66]). Knowledge, which generates new ideas and methods, plays a key role in combining capital and human capital in the process of innovation. However, it is often ambiguous and hard to measure. The previously mentioned reports [61,62] utilize the numbers of published academic papers to indicate the stock of knowledge, since they represent the new scientific findings to some extent. Therefore, the numbers of papers published by researchers in major domestic and foreign scientific and technical journals in each city are used as the incremental knowledge formed in the region, thereby measuring the level of regional knowledge input (e.g., [67,68]).

Second, factors such as educational support, communication infrastructure, and innovation service institutions in a region constitute the innovation environment of the region and indirectly contribute to the value of the regional innovation. According to the reports on China's regional innovation mentioned previously, indicators addressing the regional governments' efforts to improve the quality of human capital, reduce the cost of information, and provide services to innovators are necessary to evaluate the environment of innovation in a region [61,62]. A major way to improve the quality of human capital in a region is to support education, as it provides the training for basic skills and knowledge to enhance the overall ability to engage in innovative activities. The proportion of educational spending to the fiscal expenditure in a region reflects the intensity of educational support and thereby is selected as an indicator. The infrastructure for the Internet and telecommunications operates as the hardware fundamentals for communication and the sharing of the information related to science and technology. As technologies grow increasingly complicated and interrelated, the creation of cutting-edge technologies rely heavily on the communication and cooperation between innovators at different locations. The construction of Internet and telecom infrastructure benefits the innovators by alleviating the burden of the cost for communication. As suggested by the reports on China's regional innovation [61,62], the numbers of communication devices and Internet users can effectively describe the environmental innovation factor, which contributes to the communication between innovators. Thus, this indicator is included in the innovation indicators system in this paper. In addition, to reduce the possible endogeneity in the estimations, the indicator representing the regional government expenditure on education is converted to the proportion of educational expenditure to fiscal expenditure, and the indicator representing the intensity of Internet and telecom infrastructure is calculated as the share of communication devices and Internet users in the population [69–71]. Besides the regional educational support and communication infrastructure, the intensity of services for local innovators is also of great importance in producing innovative outputs in a region. The services for the innovators can be provided by different organizations, including incubators and patent agents. Such organizations often provide financial and informational support for local innovators to convert new ideas, projects, and inventions into final products and to help market these products. Thus, a region with a larger number of such organizations is expected to have better innovation environment. Furthermore, studies have found evidence of the positive impacts of national incubators on local innovation performance [72,73]. In addition, although few studies pay attention to the potential influence of patent agencies on local innovative activities, it may be an effective indicator of services for local innovators.

Third, the reports on China's regional innovation [61,62] also included indicators of regional economic environmental factors, which address the external conditions for

the dynamic equilibrium of the technology market influencing the decisions of both the supply and demand sides of technology and the benefits of technology formation. The per capita gross regional product (GRP) in a region was utilized to reflect the overall economic development [60–62]. Moreover, the regional economic development determines the size of the local market, which affects the value of the patents, as implied by Schankerman and Pakes [9]. Besides the regional per capita GRP, the foreign capital actually utilized is considered as another key factor determining the outputs of innovation in a region. The underlying logic is that foreign investment often leads to the inflow of advanced technologies from foreign countries, and through the spillover effect, such technologies would be quickly diffused, allowing domestic producers to imitate and improve the level of technology [74,75]. The development of finance in a region also determines the innovation outputs, since designing new products, improving production lines, and other forms of innovative efforts require large amounts of investment, especially when high-value projects developing cutting-edge technologies are involved. The ratio of the total market value of the listed enterprises to the GRP is selected as the indicator of the development of regional finance, as suggested in the "Evaluation Report on China's Regional Innovation Capacity 2021" [61]. In addition to these indicators of regional economic environment mentioned above, the average employee wage is included in the indicator system to reflect the level of household income in a region. Since 9.52% of the invention patents in the sample are applied for by individual households, it is significant to consider the potential effect of a household's income on its innovative activities. According to the componential theory of creativity [76], the theory of planned behavior [77], and the theory of individual innovation [78], individuals' personal competences, which can be reflected by their incomes, may be the key components to encourage individual innovative activities [79]. Further, prosperous individuals tend to pursue different life goals other than subsistence and safety, as implied in Maslow's hierarchy of needs [80]. Thus, Chen et al. [81] argued that prosperous individuals are more likely to conduct innovation activities. Studies have confirmed the positive impact of income on household innovation [81–83]. Therefore, this paper also includes the average wage of an employee in the indicator system to address the potential effect of income on regional innovation.

The estimation of the weighted value of each tier 1 indicator in each sample year is conducted by the entropy weight method. The city level is selected as the benchmark regional administrative division in establishing the regional innovation indicator system. The comprehensive measure of each tier 1 indicator is calculated based on the values of the four associated tier 2 indicators and their corresponding weights estimated using the entropy weight method based on the following equations:

$$Y_{ij} = \frac{X_{ij} - X_{j,min}}{X_{j,max} - X_{j,min}}, \tag{15}$$

$$e_j = -k \sum_{i=1}^{n} p_{ij} \cdot \ln p_{ij} \ , \ p_{ij} = Y_{ij} / \sum_{i=1}^{n} Y_{ij}, \tag{16}$$

$$F_i^h = \sum_{j=1}^{4} w_j^h X_{ij}^h \ , \ w_j = (1 - e_j) / \sum_{j=1}^{m} (1 - e_j). \tag{17}$$

where Equation (15) shows the process of normalization of the data, $Y$ represents the normalized dimensionless data, $X$ represents the data in the $i$th city and $j$th tier 2 indicator in a sample year, and $X_{j,min}$ and $X_{j,max}$ represent the minimum and maximum of the data for the $j$th tier 2 indicator in the sample year, respectively. Equation (16) calculates the entropy of the $j$th tier 2 indicator, wherein $k = 1/\ln n$, with $n$ being the number of cities sampled in the corresponding year. The calculation of the three tier 1 indicators is presented in Equation (17), where $F_i^h$ represents the $h$th tier 1 indicator for the $i$th city in a sample year and $w_j^h$ represents the entropy weight for $j$th tier 2 indicator of the $h$th tier 1 indicator. A similar calculation is conducted for each year throughout the sampling time span.

### 3.3. Data Description

This study uses two datasets: one is the micro-level data for patents, and the other is the macro-level data for each (tier 2) indicator in the regional innovation indicator system.

### 3.3.1. Patent Data

(1) Data sources

A total of 2,799,639 invention patents issued by patent offices and applied for by domestic applicants in 31 Chinese provinces (patents applied for in Hong Kong, Macau, and Taiwan were not included because they have their own patent offices) were collected from the Patent Information Search Platform (Available at http://search.cnipr.com (accessed on 31 May 2022)), which was established by China's statutory publisher of patent literature. The date range for the collected patent applications was 1 January 1987 to 31 December 2019. The earliest patent applications in China date to 1 April 1985, when the first amendment of the Patent Law of the People's Republic of China was published. At that time, the statutory renewal limit was 15 years, which was later changed to 20 years for those who applied after 1 January 1993, when the third amendment of the Patent Law went into effect. Then, on 10 December 2001, the CNIPA extended the statutory renewal limit to 20 years for patents applied for before 1 January 1993. Therefore, given these differences in statutory renewal limits, this study excluded patents applied for before 1 January 1993 (those patents had a 15-year statutory renewal limit and had already expired before the extension). The data for the invention patents were collected in 2022. However, it takes about three years on average to grant approval for an invention patent after its application in China, and the renewal information for the invention patents applied for after 2019 was mostly unobservable, meaning it could not be utilized for the estimation of the innovation value. Therefore, the invention patents applied for after 2019 were excluded from the data set. A small number of invention patents that were still in the examination stage or missing information about the termination date of patent rights protections at the end of 2020 were excluded. Among the collected patents, 609,758 had terminated there protection by the end of 2020. In addition to the patent information, the schedule of the patent renewal fees was collected from the China National Intellectual Property Administration (CNIPA) and converted to real terms using GDP deflators (the base year was 2010).

(2) Patent cohorts

The cohort of a patent is identified based on the year and city of application, the type of applicant, and the industrial classification for the national economic activity of the patent. The data contain the IPC codes for the patents, which identify the technical area. However, IPC codes might not directly relate to regional economic activity. Therefore, the IPC codes are converted to their associated industrial classifications based on the "Reference Table for IPC and Industrial Classification for National Economic Activities (2018)" published by CNIPA, (available at https://www.cnipa.gov.cn/art/2018/10/8/art_75_131968.html (accessed on 31 May 2022)) (henceforth, "industrial sector". The dataset contains patents covering 33 years (1987–2019), 282 cities (including municipalities and prefecture-level cities), four types of applicants (individuals, nonlisted enterprises, listed enterprises, and other public organizations, including government departments, government-affiliated institutions, colleges, and research institutions), and eight industrial sectors (agriculture, forestry, livestock, and fisheries; mining; manufacturing; electricity, heat, gas, and water production and supply; construction; information transmission, software, and information; technical services; and health and social work). Altogether, these categorical variables form 53,056 cohorts in the patent dataset.

(3) Renewal proportion in the sample

The renewal proportion for patents in a cohort at a specific age is calculated based on the ratio of the number of patents remaining valid at that age to the total number of patents in the cohort. The patent age is not directly provided but can be calculated based on the difference between the date the rights expired and the date of application, which is then rounded down to the nearest integer as its age. Thus, up to the age of a patent, it keeps

renewing and remains valid. The average age of the 609,758 expired patents in the sample was 6.60 years (standard deviation: 0.004 years). Table 2 shows the number of patents and the renewal proportions (at the end of the 5th, 10th, 15th, and 20th years after application) grouped by year of application, type of applicant, and industrial sector. Overall, the renewal proportions in the 10th, 15th, and 20th years increase in the application year, while the renewal proportions in the 5th year are somewhat ambiguous. Considering the time span of renewals for the majority of expired patents is relatively short (73% of total sample patents renewed up to the fifth year), it cannot be concluded explicitly that the values of invention patents in China increases with time. Moreover, the patents applied for by listed firms have relatively higher renewal proportions, and some small differences can be overserved between renewal proportions in different industrial sectors, which implies differences in frequency distribution among the groups of patents. Differences in frequency distributions are verified using K–S tests. Among the 562 pairs of renewal frequency distributions, only seven pairs have an estimated *p*-value greater than 0.01, which indicates that most of the groups have different renewal frequency distributions. It is necessary, therefore, to control the potential heterogeneity in patent values using dummies for patents' categories.

### 3.3.2. Data for Regional Innovation Indicators

The data for the tier 2 indicators were mainly obtained from the China City Statistical Yearbooks from 1985 to 2019. Owing to data limitations, the number of scientific researchers was substituted by the number of people employed in the sector of "scientific research, technical services, and geological prospecting." In addition, data for the number of local telephone subscribers were missing for 2017, while data for mobile phone and Internet service subscribers were missing for 2000 and before. To simplify the indicator, the number of subscribers for all three was summed up to calculate the ratio of subscribers to population annually. The unit of measure for the amount of foreign capital actually used per capita was converted to yuan based on the annual average exchange rate of the US dollar to the RMB each year.

The number of papers published in domestic academic journals was collected from the China National Knowledge Infrastructure (CNKI) database by matching the addresses of authors' affiliated institutions with the sample cities for each year. The number of papers published in foreign academic journals was collected from the Web of Science using the sample cities' names as the search criteria for each year. National-level incubators and their locations were obtained from the website of the Torch High Technology Industry Development Center, Ministry of Science and Technology (The lists of national-level incubators can be found at http://www.chinatorch.gov.cn (accessed on 31 May 2022)). Information about patent agencies, including addresses and years of establishment, was collected from the CNIPA website (Available at http://dlgl.cnipa.gov.cn/txnqueryAgencyOrg.do (accessed on 31 May 2022)). Lastly, the total market value of listed enterprises is the sum of the total annual individual market value of all listed A-share enterprises in each city of registration.

All monetary data in the innovation indicator system are deflated by the GDP deflator to reduce the influence of macro-level prices. (The data for GDP deflators are not directly provided by any official of statistics in China. However, as suggested by Shen and Wang [84], the GDP deflators can be calculated based on the nominal GDP and GDP at a constant price, which are provided by the National Bureau of Statistics of China (the base year is 2010)). After excluding cities missing considerable data, data for 282 sample cities were retained. In the remaining data, missing values were filled in using regression interpolation. Finally, the values of the tier 2 indicators were calculated according to their definition and then used to calculate the indexes of the three tier 1 indicators. Table 3 presents the descriptive statistics of all tier 2 indicators.

**Table 2.** Number of invention patents and proportion of renewals for the full sample.

| Category of Sample Patents | Total Number of Patents in the Cohort | Renewal Proportion (%) | | | | Percentage of Unexpired Patents (%) |
|---|---|---|---|---|---|---|
| | | 5th Year | 10th Year | 15th Year | 20th Year | |
| Whole Sample [1] | 2,799,639 | 73 | 16 | 2 | 1 | 78 |
| Year of Application | | | | | | |
| 1987 | 1447 | 92 | 25 | 7 | 2 | 0 |
| 1988 | 1589 | 90 | 25 | 7 | 2 | 0 |
| 1989 | 1436 | 87 | 26 | 7 | 3 | 0 |
| 1990 | 1652 | 88 | 27 | 8 | 4 | 0 |
| 1991 | 1857 | 86 | 30 | 8 | 3 | 0 |
| 1992 | 2394 | 88 | 36 | 11 | 6 | 0 |
| 1993 | 2711 | 95 | 43 | 13 | 8 | 0 |
| 1994 | 2738 | 98 | 39 | 15 | 9 | 0 |
| 1995 | 2808 | 99 | 35 | 19 | 10 | 0 |
| 1996 | 3474 | 96 | 32 | 18 | 11 | 0 |
| 1997 | 4020 | 92 | 32 | 19 | 12 | 0 |
| 1998 | 4817 | 92 | 34 | 22 | 12 | 0 |
| 1999 | 6051 | 91 | 38 | 25 | 12 | 0 |
| 2000 | 8670 | 89 | 46 | 26 | 13 | 0 |
| 2001 | 11,425 | 90 | 50 | 29 | | 16 |
| 2002 | 18,571 | 88 | 53 | 32 | | 22 |
| 2003 | 26,328 | 88 | 56 | 30 | | 23 |
| 2004 | 30,768 | 88 | 56 | 29 | | 26 |
| 2005 | 41,574 | 94 | 59 | 32 | | 31 |
| 2006 | 52,852 | 95 | 61 | | | 37 |
| 2007 | 64,338 | 96 | 62 | | | 41 |
| 2008 | 82,625 | 96 | 60 | | | 48 |
| 2009 | 106,456 | 94 | 57 | | | 53 |
| 2010 | 127,230 | 93 | 60 | | | 60 |
| 2011 | 161,289 | 93 | | | | 69 |
| 2012 | 214,565 | 94 | | | | 76 |
| 2013 | 280,558 | 91 | | | | 80 |
| 2014 | 310,103 | 90 | | | | 85 |
| 2015 | 325,582 | 92 | | | | 91 |
| 2016 | 352,744 | | | | | 95 |
| 2017 [2] | 270,453 | | | | | 99 |
| 2018 [2] | 192,387 | | | | | 99 |
| 2019 [2] | 84,127 | | | | | 99 |
| Type of Applicant [1] | | | | | | |
| Individual | 266,419 | 74 | 18 | 3 | 1 | 51 |
| Nonlisted firms | 1,425,615 | 77 | 21 | 3 | 1 | 87 |
| Listed firms | 189,437 | 96 | 54 | 6 | 2 | 90 |
| Public sector | 918,168 | 67 | 9 | 1 | 0 | 69 |
| Industrial Sectors [1] | | | | | | |
| Agriculture, forestry, livestock, and fisheries | 40,565 | 60 | 8 | 1 | 0 | 65 |
| Mining | 10,322 | 72 | 16 | 3 | 1 | 80 |
| Manufacturing | 2,367,409 | 73 | 16 | 3 | 1 | 78 |
| Electricity, heat, gas, and water production and supply | 5451 | 70 | 10 | 1 | 0 | 76 |
| Construction | 47,148 | 64 | 11 | 2 | 1 | 82 |
| Information transmission, software, and information | 84,106 | 79 | 16 | 2 | 0 | 86 |
| Technical services | 16,055 | 84 | 21 | 0 | 0 | 86 |
| Health and social work | 228,583 | 75 | 16 | 1 | 0 | 83 |

Note [1]: The corresponding renewal ratios are the proportion of the conditions for the renewal of lapsed invention patents in each cycle. Note [2]: Some invention patents filed from 2017 to 2019 are still under reexamination and are not included in the sample; hence, the sample of granted invention patents is small.

## 4. Results and Discussion

### 4.1. Estimation of Regional Innovation Value Model

4.1.1. Correlation between Regional Innovation Indicators and Sample Renewal Proportion

Before estimating the regional innovation value model, a graphic illustration is presented (Figure 1) of the correlation between the log indicators of the regional innovation inputs ($\ln Inp_{i,L2}$), innovation environment ($\ln Env_{i,L2}$), and economic environment ($\ln Eco_{i,L2}$),

as well as the logs of the sample renewal proportions ($\ln P_{it}$), to preview the indicators' potential influence on the patent value.

**Table 3.** Descriptive statistics for regional innovation indicators.

| Regional Innovation Indicators | Mean | Std. Dev. | Min | Max |
|---|---|---|---|---|
| Innovation Input | | | | |
| Number of R&D personnel per 10,000 people | 15.04 | 0.29 | 0.22 | 524.42 |
| Government R&D spending as proportion of fiscal expenditure | 0.01 | 0.011 | 0.00 | 0.21 |
| Number of papers published in domestic academic journals per 10,000 people | 5.69 | 10.339 | 0.00 | 122.40 |
| Number of papers published in foreign academic journals per 10,000 people | 0.51 | 3.469 | 0.00 | 103.22 |
| Innovation Environment | | | | |
| Government educational spending as a proportion of fiscal expenditure | 0.20 | 0.06 | 0.02 | 0.49 |
| Ratio of telephone, mobile phone, and Internet subscribers to population | 0.59 | 0.93 | 0.00 | 13.53 |
| Number of national-level incubators for innovative enterprises | 0.82 | 3.27 | 0.00 | 61.00 |
| Number of patent agencies | 2.01 | 14.94 | 0.00 | 637.00 |
| Economic Environment | | | | |
| GRP per capita | 23,187.09 | 32,770.05 | 426.35 | 455,125.43 |
| Average wage of employees | 21,733.51 | 17,954.21 | 14.04 | 169,823.84 |
| Amount of foreign capital actually used per capita | 689.04 | 1807.53 | 0.04 | 30,415.53 |
| Ratio of total market value of listed enterprises to GRP | 0.16 | 0.38 | 0.00 | 7.69 |

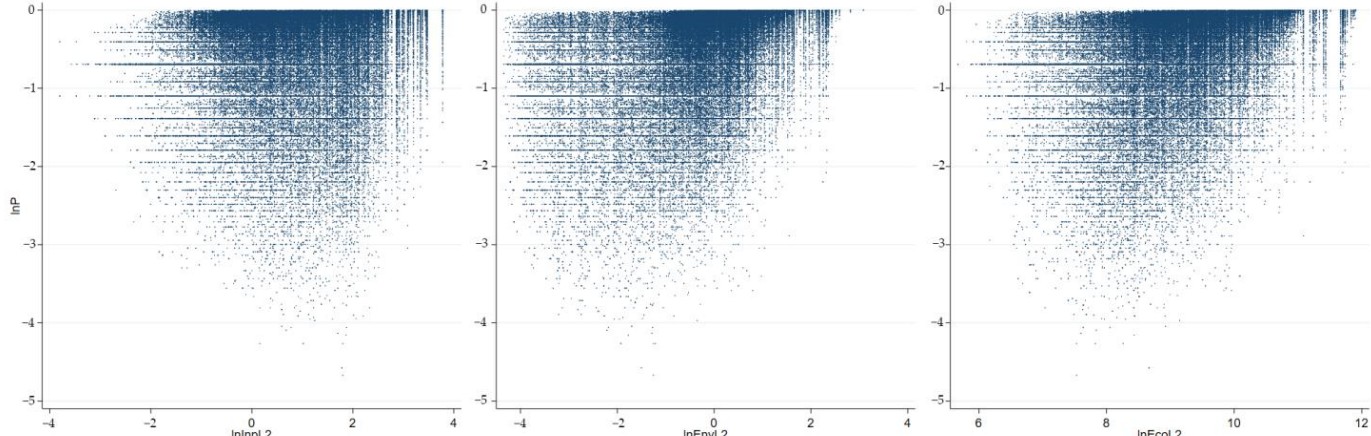

**Figure 1.** Scatterplot of the correlation between the logs of regional innovation indicators and renewal proportions.

As seen in Figure 1, the logs of the regional innovation indicators and of the sample renewal proportions show a somewhat skewed, inverted U-shaped relationship. Considering the change in the scale of variables owing to the log transformation, the inverted U-shaped relationship implies a potential lognormal distribution of patent renewal across regions (cities) with various levels of regional innovation indicators. This provides indirect evidence that the regional innovation indicators have a significant influence on the patent value; that is, the coefficients of elasticities for the indicators in the regional innovation value model are possibly statistically significant.

4.1.2. Regression Results for the Regional Innovation Value Model

The regional innovation value models established in Equations (10), (11) and (14), along with the original patent renewal model in Equation (5) (for comparison), are estimated using the nonlinear least-squares method. Table 4 presents the results.

**Table 4.** Results of the regional innovation value models.

| Parameters | Regional Innovation Value Model | | | Patent Renewal Model |
|---|---|---|---|---|
| | (I) | (II) | (III) | |
| $\mu_{i0}$ [1] | 10.195 *** | 9.354 *** | 9.973 *** | 10.330 *** |
| | (0.075) | (0.063) | (0.085) | (0.045) |
| $b_1$ | 0.090 *** | 0.015 *** | 0.017 *** | |
| | (0.003) | (0.003) | (0.004) | |
| $b_2$ | −0.055 *** | 0.026 *** | 0.028 *** | |
| | (0.005) | (0.006) | (0.007) | |
| $b_3$ | 0.084 *** | 0.023 *** | 0.026 *** | |
| | (0.006) | (0.006) | (0.006) | |
| $\sigma$ | 0.613 *** | 0.473 *** | 0.525 *** | 0.470 *** |
| | (0.008) | (0.005) | (0.007) | (0.005) |
| $d$ | 0.744 *** | 0.813 *** | 0.757 *** | 0.814 *** |
| | (0.004) | (0.003) | (0.005) | (0.003) |
| $\theta_1$ | | | −0.029 *** | |
| | | | (0.004) | |
| $\theta_2$ | | | −0.048 *** | |
| | | | (0.003) | |
| Application year | | Control | Control | Control [2] |
| Application city | | | | Control |
| Applicant type | | Control | Control | |
| Industrial sector | | Control | Control | |
| N | 503,185 | 503,185 | 503,185 | 503,185 |
| Adjusted $R^2$ | 0.544 | 0.571 | 0.571 | 0.554 |
| WSSE | 0.164 | 0.156 | 0.155 | 0.163 |

Note: *** denotes parameter estimates statistically significant at the 0.01 level of significance, and values in parentheses are standard errors. Note [1]: $\mu_{i0}$ is the mean value of the initial distribution of patent proceeds from inventions filed by individuals in 1987 in agriculture, forestry, livestock, and fisheries. Note [2]: Owing to the large number of urban dummy variables, the econometric software (Stata17) was unable to estimate a model that included both application year and urban dummy variables, so a quadratic polynomial about the application year was used to control for its effect on the mean of the initial distribution of returns, i.e., $\mu_i = \mu_{i0} + \alpha_1$ (application year–1987) $+ \alpha_2$ (application year–1987)$^2 + \beta' x_i$.

In Table 4, the regional innovation value models (I), (II), and (III) refer to Equations (10), (11), and (14), respectively. Model (I) only introduces the tier 1 regional innovation indicators; model (II) further controls for the application year, applicant type, and industrial sector; model (III) allows for some variation in the decay rates of patent returns. The last column shows the results of the original patent renewal model as described in Equation (5). Following Amemiya [85] and Schankerman and Pakes [9], the weighted sum of the squared error (WSSE) is calculated based on the sum of the squared difference between the sample and the fitted renewal proportions weighted by the binomial sampling variance of renewal proportions (i.e., $P_{it}(1 − P_{it})/N_i$, where $N_i$ is the number of patents on cohort $i$). This gives a general view of the overall fitness of the models.

As shown in Table 4, the larger value for the adjusted $R^2$ and smaller value for the WSSE in model (III) indicate the good overall fit of the regional innovation value models to the sample data. In addition, comparing the values of the adjusted $R^2$ and the WSSE in models (III) and (I), it can be inferred that the controls for the categorical variables of sample patents do increase the overall explanatory power of the model.

The estimates for elasticity parameters $b_1$, $b_2$, and $b_3$, which are the focus of this study, are statistically significant and generally have the expected positive sign, thereby rejecting the original hypothesis that the elasticity coefficients of the regional innovation indicators are zero. This indicates that the regional innovation indicators do have a significant influence on the value of patents in the associated regions. However, the estimated elasticity of the innovation environment indicator, ln$Env_{i,L2}$, in model (I) is significantly negative, while in models (II) and (III), the estimates are significantly positive. This result, along with the increase in the adjusted $R^2$ and the decrease in WSSE in models (II) and (III),

implies that other categorical variables controlled by dummy variables might reduce the endogeneity caused by some omitted variables.

The estimates of $\mu_{i0}$ are generally smaller in the regional innovation value model (10.195, 9.354, and 9.973 in models (I), (II), and (III), respectively) compared with the estimate in the patent renewal model (10.330). This implies a potential overestimation of the patent value and regional innovation value by the patent renewal model. Comparing all four models, the estimates for $\sigma$ are somewhat ambiguous, indicating that the variation in the patents' initial returns might differ using models with different setups. However, the differences are not quite notable and do not significantly affect the calculation of the regional innovation value. In model (III), where the decay rates of patent returns are separately estimated, the results indicate that the decay rates decrease in actual years, or equivalently the patent value depreciates faster in more recent years. This could be caused by decreases in renewal fees in China in more recent years.

In general, the regional innovation value model fits the Chinese invention patent samples better based on the empirical results. The statistical significance of the estimated elasticity of the regional innovation input, innovation environment, and economic environment indicators validates the model's effectiveness and confirms the first hypothesis (H1) proposed in the previous section. Comparing the estimated $\mu_{i0}$ indicates that the patent renewal model might overestimate the patent value and regional innovation value.

### 4.1.3. Robustness Test

Despite the large number of observations in the patent sample, given that the nonlinear model has more stringent assumptions for the sample data, coupled with the large number of cohorts in the sample, it is necessary to conduct a robustness test on the regional innovation value model. The robustness test is conducted by fitting the model with subsamples of different application years, specifically those that were applied for 1987 to 1995, 1987 to 2005, and 1987 to 2015. Table 5 presents the regression results.

**Table 5.** Robustness test of the regional innovation value model.

| Parameters | Application Years 1987–1995 | Application Years 1987–2005 | Application Years 1987–2015 | Full Sample |
|---|---|---|---|---|
| $\mu_0$ | 9.326 *** | 8.929 *** | 9.975 *** | 9.973 *** |
| | (0.117) | (0.086) | (0.089) | (0.085) |
| $b_1$ | 0.059 *** | 0.039 *** | 0.020 *** | 0.017 *** |
| | (0.005) | (0.004) | (0.004) | (0.004) |
| $b_2$ | 0.072 *** | 0.066 *** | 0.036 *** | 0.028 *** |
| | (0.012) | (0.008) | (0.007) | (0.007) |
| $b_3$ | 0.012 | 0.065 *** | 0.027 *** | 0.026 *** |
| | (0.010) | (0.007) | (0.007) | (0.006) |
| $\sigma$ | 0.387 *** | 0.444 *** | 0.526 *** | 0.525 *** |
| | (0.008) | (0.007) | (0.007) | (0.007) |
| $d$ | 0.844 *** | 0.831 *** | 0.756 *** | 0.757 *** |
| | (0.007) | (0.006) | (0.006) | (0.005) |
| $\theta_1$ | −0.047 *** | 0.035 *** | −0.029 *** | −0.029 *** |
| | (0.004) | (0.003) | (0.004) | (0.004) |
| $\theta_2$ | −0.053 *** | −0.072 *** | −0.049 *** | −0.048 *** |
| | (0.010) | (0.004) | (0.003) | (0.003) |
| Application Year | Control | Control | Control | Control |
| Applicant Type | Control | Control | Control | Control |
| Industrial Sector | Control | Control | Control | Control |
| N | 83,480 | 250,729 | 472,345 | 503,185 |
| Corrected $R^2$ | 0.601 | 0.537 | 0.558 | 0.571 |
| WSSE | 0.140 | 0.148 | 0.155 | 0.155 |

Note: *** denotes parameter estimates that are statistically significant at the 0.01, 0.05 significance level, and values in parentheses are standard errors.

Overall, the results demonstrate the robustness of the results of the regional innovation value models. After adjusting for the application year, the overall fitness of the models does not vary much, and almost all estimates remain statistically significant. The absolute value

of the estimated parameter $\mu_{i0}$ tends to first decrease and then increase with the expansion of the time window, which is consistent with the trending of the estimated coefficients of the dummy variables for the application years in all models. The estimated coefficients of the application year dummies (1988–2019) in model (III) are 0.123, 0.087, −0.012, −0.013, 0.460, 0.645, 0.586, 0.374, 0.305, 0.304, 0.190, 0.215, 0.163, 0.223, 0.132, 0.122, 0.077, 0.062, 0.074, −0.004, −0.143, −0.275, −0.385, −0.404, −0.413, −0.447, −0.512, −0.478, −0.435, −0.428, −0.424, and −0.537. These represent the differences in the mean patent value in each application year relative to the mean patent value in 1987. One can infer that the mean of patent values first increases and then decreases during the years of application. Similar results are found for all models, including the original patent renewal model. The estimates for $\sigma$ and d, as well as $\theta$s, are mostly consistent with the results in model (III). The only differences are the estimates of $b_3$ in the first column and $\theta_1$ in the second column, which can be neglected because the sign of $b_3$ is as expected and the decay rates could be potentially increasing in a certain time window.

*4.2. Regional Innovation Values in China*

4.2.1. Calculation of Regional Innovation Value

Calculating the regional innovation value consists of two steps: first, calculate the values of sample patents using the estimates from the previous results and the Monte Carlo simulation; second, aggregate the total values of sample patents according to their application years and cities. Note that the city-level innovation value is the benchmark in this study.

Using the estimates from the regional innovation value model (III), the values of the sample patents can be calculated based on the process described in Section 2.1. The fitted parameters needed for the Monte Carlo simulation are given as follows:

$$\hat{\mu}_i = \hat{\mu}_{i0} + \hat{b}_1 \ln Inp_{i,L2} + \hat{b}_2 \ln Env_{i,L2} + \hat{b}_3 \ln Eco_{i,L2} + \hat{B}D_i, \tag{18}$$

$$\hat{\sigma}_i = \hat{\sigma}, \tag{19}$$

$$\hat{d}_{rt} = \hat{d}\exp(\hat{\theta}_1 D_{rt}^1 + \hat{\theta}_2 D_{rt}^2). \tag{20}$$

With the parameters (Equations (18) and (19)) defining the lognormal distribution of the initial returns of patents in the $i$th cohort, a set of pseudo-random numbers representing the simulated initial returns is generated. Then, following the dynamics of the return (i.e., $R_{it} = R_{i0} \cdot d^{-t}$) with the estimates for the decay rate (Equation (20)) and the renewal condition (described in Inequality (2)), the optimal renewal period ($\widetilde{T}_j^*$) (the last period that a pseudo-number satisfies the renewal condition) along with the discount values throughout the renewal period (calculated based on Equation (7)) of the $j$th pseudo-number are simulated. Lastly, the average discount value of the pseudo-numbers for each optimal renewal period ($\widetilde{T}^* = 1, 2, \ldots, 20$) are calculated as the value of a patent in the $i$th cohort and expired at each period:

$$\hat{V}_t = \frac{\sum_j \widetilde{V}_j\left(\widetilde{T}^*\right)}{\widetilde{n}_{\widetilde{T}^*}}, \ t = \widetilde{T}^*, \tag{21}$$

where $\widetilde{V}_j\left(\widetilde{T}^*\right)$ stands for the discount values throughout the renewal period of the $j$th pseudo-number and $\widetilde{n}_{\widetilde{T}^*}$ denotes the number of simulated pseudo-numbers with the optimal renewal period of $\widetilde{T}^*$. In addition, for the sample patents that are not expired, their values are simulated in the same way but the expected number of these patents that will be expired in future periods ($\widetilde{n}_{\widetilde{T}^*}^u$) is calculated based on the simulated proportions of renewals.

Once the values of the sample patents are calculated, the summation of the patent value for all cohorts across the application year and city is used as the estimate of a city's innovation value in the corresponding year. Thus, it follows that:

$$City\ Innovation\ Value = \sum_{i} \sum_{t=1}^{20} \left(n_{i,t}^{e} + \tilde{n}_{i,t}^{u}\right) \cdot \hat{V}_{i,t}, \qquad (22)$$

where $n_{i,t}^{e}$ is the number of expired patents at age $t$ in cohort $i$ in the addressed year and city, $\tilde{n}_{i,t}^{u}$ is the simulated number of unexpired patents, and $\hat{V}_{i,t}$ the estimated value of the patents at age $t$ in cohort $i$ (as in Equation (21)). In addition to the benchmark (city-level innovation value), the innovation values at the province, area, and nationwide levels can be calculated in the same manner.

### 4.2.2. Nationwide Innovation Value

Using the above method, the nationwide innovation value as well as the average values of all sample patents are calculated for each sample year. Figure 2 presents the nationwide innovation value and the average innovation value (i.e., the average of the total patent value in each year) from 1987 to 2019.

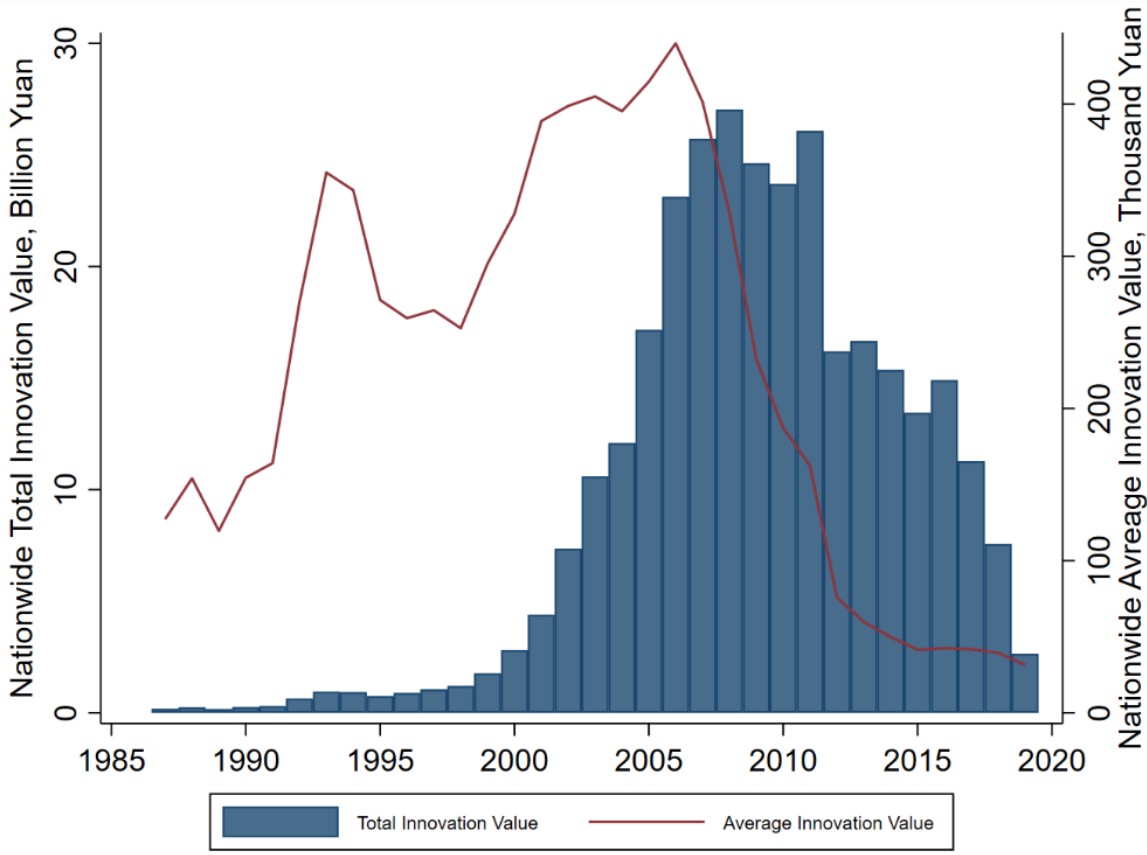

**Figure 2.** Nationwide total and average innovation values.

It is noteworthy in Figure 2 that both the total value and average innovation value rise first and then decline in recent years. This finding is consistent with the estimated coefficients of the dummies representing the application year in the results of model (III). Although the number of invention patents issued kept increasing for most of the year (see Table 2), this finding provides evidence that China's overall innovation output is large in quantity but low in quality, which indicates the unsustainable development of nationwide innovation, particularly after 2008, when both the total and average values of innovation start to decline. However, this finding could be misleading. First, the decline in the total and average innovation values started around 2008, while both were increasing prior to 2008. Thus, for a longer period, the innovation values (as well as the amount of innovation output) continued to increase. Second, the agglomeration of innovation resources might cause a concentration of innovation values in specific regions, which could

result in increases in both the quality and quantity of innovation output in those regions. Therefore, further investigation of regional innovation value is required.

### 4.2.3. City-Level Innovation Value

The innovation value of all sample cities from 1987 to 2019 is calculated based on the previous discussion. Table 6 presents the total innovation value, average patent value, and patent count in cities whose total innovation values ranked in the top and bottom 10 of all sample cities in 1987, 2000, 2010, and 2019. Overall, the average total city-level innovation value changes from 1.072 million yuan in 1987 to a peak of 37.503 million yuan in 2010, followed by a decrease to 9.514 million yuan in 2019, exhibiting a trend of increasing first and then decreasing. A similar trend can be observed in terms of the average values of all patents nationwide in these years (the second cell in the last row in each year).

Regarding the rankings of city-level innovation values in the four sample years, it is clear that the innovation value of the ones at the very top is considerably different from that of the others at the top. Such differences are even greater compared with the bottom-ranked cities, indicating a severe imbalance in the spatial distribution of regional innovation values. In addition, only Beijing, Shanghai, and Chengdu are consistently listed at the top, and Beijing always has the highest innovation value. Tianjin, Wuhan, Guangzhou, and Shenzhen are also regulars at the top of the list. The consistency of these cities in terms of the innovation values implies that innovations value are potentially concentrated in the regions anchored by these cities. Meanwhile, there is no notable pattern for cities in the bottom 10. However, the calculated city-level innovation value for all sample cities behaves in the manner of a skewed long-tailed distribution. In other words, most cities have minor innovation value while a small number of cities have the most innovation value.

The situation is quite different in terms of the average value of patents in each city, especially in 2010 and 2019. The average patent value in the bottom 10 cities during those two years is much higher than in 1987 and 2000, and the differences across all cities are not very large after 2010. This implies that the overall value of the innovation output has increased nationwide, especially in cities whose total innovation value is much smaller than that of the top cities, and is distributed more evenly in recent years.

In general, it can be inferred that only a small number of cities achieve relatively sustainable innovation development, as these cities' total innovation values remain relatively high and increase in most years. Although the average value of innovation in each city keeps rising notably in most years, the total innovation values for the majority of cities, especially the bottom ranked cities, remain at low levels, which indicates that the majority of cities are unable to sustain the continuous improvement of the quality of technological innovation.

### 4.2.4. Area-Level Innovation Value

The total innovation value of China's four major areas (i.e., Eastern, Central, Western, and Northeast China; Eastern China includes Beijing, Tianjin, Hebei, Shanghai, Jiangsu, Zhejiang, Fujian, Shandong, Guandong, and Hainan; Central China includes Shanxi, Anhui, Jiangxi, Henan, Hubei, and Hunan; Western China includes Inner Mongolia, Guangxi, Chongqin, Sichuang, Guizhou, Yunnan, Tibet, Shaanxi, Gansu, Qinghai, and Xinjiang; Northeast China includes Liaoning, Jilin, and Heilongjiang) in each year is calculated to provide a preliminary analysis of the spatial distribution of innovation values (Figure 3).

In general, the innovation value is notably higher in Eastern China than in the other areas. Central and Northeast China's innovation values are relatively close before 2009 but Central China's innovation value increases afterward. The innovation value of Western China remains the lowest over the sample period. The trends in innovation value in all four areas are similar; all increase prior to 2008 and decline afterward.

**Table 6.** Total innovation values, average patent values, and numbers of patents in the top and bottom 10 cities.

| City | Total Value [1] | 1987 Average Value [2] | Patent Count | City | Total Value [1] | 2000 Average Value [2] | Patent Count | City | Total Value [1] | 2010 Average Value [2] | Patent Count | City | Total Value [1] | 2019 [3] Average Value [2] | Patent Count |
|---|---|---|---|---|---|---|---|---|---|---|---|---|---|---|---|
| (Top 10) | | | | | | | | | | | | | | | |
| Beijing | 43.047 | 156.534 | 275 | Beijing | 880.376 | 475.109 | 1853 | Beijing | 4447.088 | 212.150 | 20,962 | Beijing | 339.010 | 31.239 | 10,852 |
| Shanghai | 23.029 | 195.159 | 118 | Shenzhen | 434.092 | 885.903 | 490 | Shenzhen | 3237.263 | 253.446 | 12,773 | Shenzhen | 136.491 | 45.695 | 2987 |
| Tianjin | 7.398 | 117.423 | 63 | Shanghai | 307.806 | 386.691 | 796 | Shanghai | 2259.849 | 203.553 | 11,102 | Zhuhai | 125.842 | 83.174 | 1513 |
| Xi'an | 5.089 | 99.792 | 51 | Chengdu | 54.503 | 283.869 | 192 | Hangzhou | 816.793 | 167.857 | 4866 | Shanghai | 118.432 | 34.338 | 3449 |
| Wuhan | 4.830 | 74.304 | 65 | Nanjing | 52.611 | 256.640 | 205 | Suzhou | 779.760 | 209.051 | 3730 | Hangzhou | 114.926 | 28.645 | 4012 |
| Guangzhou | 3.656 | 130.573 | 28 | Guangzhou | 52.460 | 200.231 | 262 | Guangzhou | 681.848 | 187.013 | 3646 | Nanjing | 99.734 | 29.763 | 3351 |
| Changchun | 3.654 | 87.000 | 42 | Suzhou | 45.023 | 372.088 | 121 | Nanjing | 585.924 | 141.801 | 4132 | Wuhan | 95.802 | 28.487 | 3363 |
| Kunming | 3.582 | 132.664 | 27 | Wuhan | 42.881 | 180.931 | 237 | Wuxi | 497.617 | 198.017 | 2513 | Guangzhou | 87.470 | 33.694 | 2596 |
| Chengdu | 3.341 | 128.498 | 26 | Tianjin | 42.058 | 200.275 | 210 | Tianjin | 488.531 | 174.725 | 2796 | Suzhou | 65.593 | 40.766 | 1609 |
| Shenyang | 3.258 | 66.496 | 49 | Hangzhou | 41.008 | 262.871 | 156 | Chengdu | 440.580 | 162.997 | 2703 | Chengdu | 61.501 | 24.328 | 2528 |
| (Bottom 10) | | | | | | | | | | | | | | | |
| Anyang | 0.005 | 4.824 | 1 | Ji'an | 0.023 | 11.743 | 2 | Liupanshui | 0.650 | 108.322 | 6 | Guangyuan | 0.047 | 15.573 | 3 |
| Qinhuangdao | 0.005 | 4.823 | 1 | Anqing | 0.019 | 9.643 | 2 | Bazhong | 0.620 | 155.021 | 4 | Songyuan | 0.037 | 36.718 | 1 |
| Zaozhuang | 0.005 | 4.819 | 1 | Jixi | 0.015 | 7.697 | 2 | Yichun | 0.602 | 150.454 | 4 | Lijiang | 0.037 | 36.535 | 1 |
| Zhoukou | 0.005 | 4.818 | 1 | Wuzhong | 0.012 | 11.826 | 1 | Wuhai | 0.542 | 90.315 | 6 | Qitaihe | 0.037 | 36.522 | 1 |
| Hebi | 0.005 | 4.816 | 1 | Jinchang | 0.012 | 11.621 | 1 | Tongchuan | 0.502 | 251.113 | 2 | Baoshan | 0.036 | 35.914 | 1 |
| Zhangzhou | 0.003 | 3.393 | 1 | Chongzuo | 0.012 | 11.614 | 1 | Jiayuguan | 0.464 | 115.906 | 4 | Guangan | 0.028 | 14.220 | 2 |
| Baishan | 0.003 | 3.362 | 1 | Chizhou | 0.012 | 11.607 | 1 | Fangchenggang | 0.403 | 134.370 | 3 | Wuwei | 0.025 | 12.630 | 2 |
| Linfen | 0.003 | 3.349 | 1 | Baiyin | 0.012 | 11.599 | 1 | Chongzuo | 0.372 | 123.833 | 3 | Longnan | 0.016 | 15.965 | 1 |
| Xinyang | 0.003 | 3.340 | 1 | Tongling | 0.008 | 7.697 | 1 | Qitaihe | 0.170 | 42.481 | 4 | Pingliang | 0.012 | 12.451 | 1 |
| Zunyi | 0.003 | 3.307 | 1 | Xuancheng | 0.008 | 7.693 | 1 | Hegang | 0.008 | 8.405 | 1 | Zhaotong | 0.012 | 12.287 | 1 |
| Average of all Sample Cities | 1.072 | 127.201 | 8 | | 10.815 | 327.895 | 33 | | 37.503 | 111.990 | 335 | | 9.514 | 31.523 | 302 |

Note [1]: The total innovation value is measured in million yuan (in real terms). Note [2]: The average innovation value is measure in thousand yuan (in real terms). Note [3]: Some invention patents applied in 2017 to 2019 were still under examination up to the time of data collection, and these patents were not included in the sample, which may cause underestimation of the innovation values.

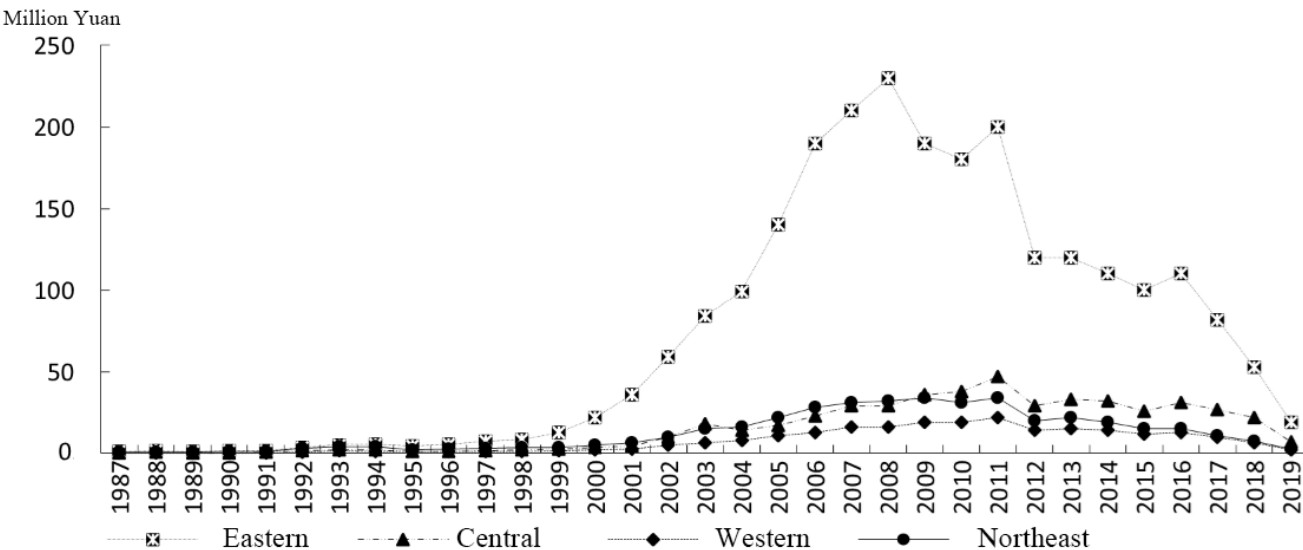

**Figure 3.** Total innovation values in China's four major regions.

The gap in innovation value between Eastern China and the other three areas also shows a trend of first widening and then narrowing. This gap is not initially notable from 1987 to 1998. It then gradually widens from 1998 to 2008 and reaches its peak in 2008, after which it begins to narrow. This narrowing of the gap in more recent years is not caused by increases in innovation in the other areas but by a decline in innovation value in Eastern China. In general, the value of innovation in the eastern region is significantly higher than that in the other three regions, while the differences in innovation value in the other three areas are not significant. This might be caused by different endowments of innovation inputs such as human capital and different industrial structures between these four areas, since both factors can significantly affect the efficiency of the technological innovation [86,87].

In general, the preliminary analysis of the innovation values at the city and area levels shows some variations in the spatial distribution of regional innovation values. The findings for city-level innovation suggest a potential agglomeration of innovation values in certain regions. The findings for area-level innovation, meanwhile, provide evidence for spatial differences and their temporal dynamics in regional innovation values. These findings suggest differences in the sustainability of regional innovation and indicate a need to further analyze the spatial distribution of regional innovation values in China.

*4.3. Spatial Distribution of Regional Innovation Values in China*

4.3.1. Overall View of the Spatial Distribution of Regional Innovation Values

To illustrate the overall spatial distribution of regional innovation values in China and the temporal dynamics, this study first calculates the average of each city's innovation values every 5 years. Then, using the natural breaks classification method, the 5-year-averages of city-level innovation values are sorted into five categories based on the ranges of innovation values. Lastly, the spatial distribution of the 5-year averages of city-level innovation values are illustrated using ArcGIS 10.8. Figure 4 presents the results.

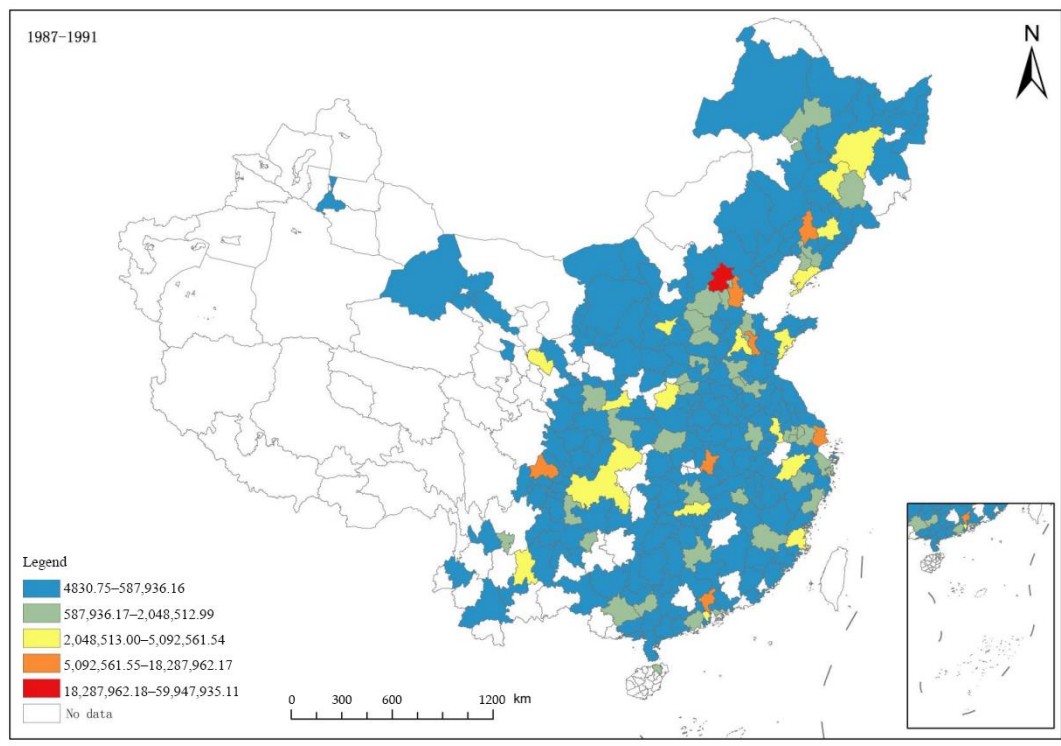

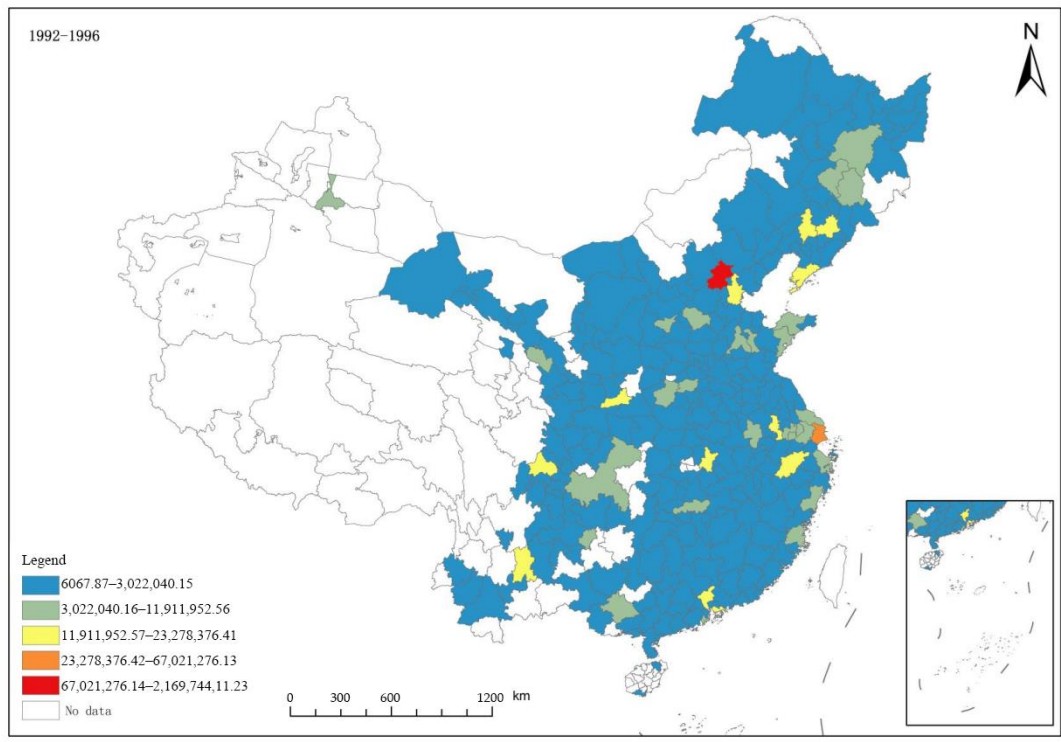

**Figure 4.** *Cont.*

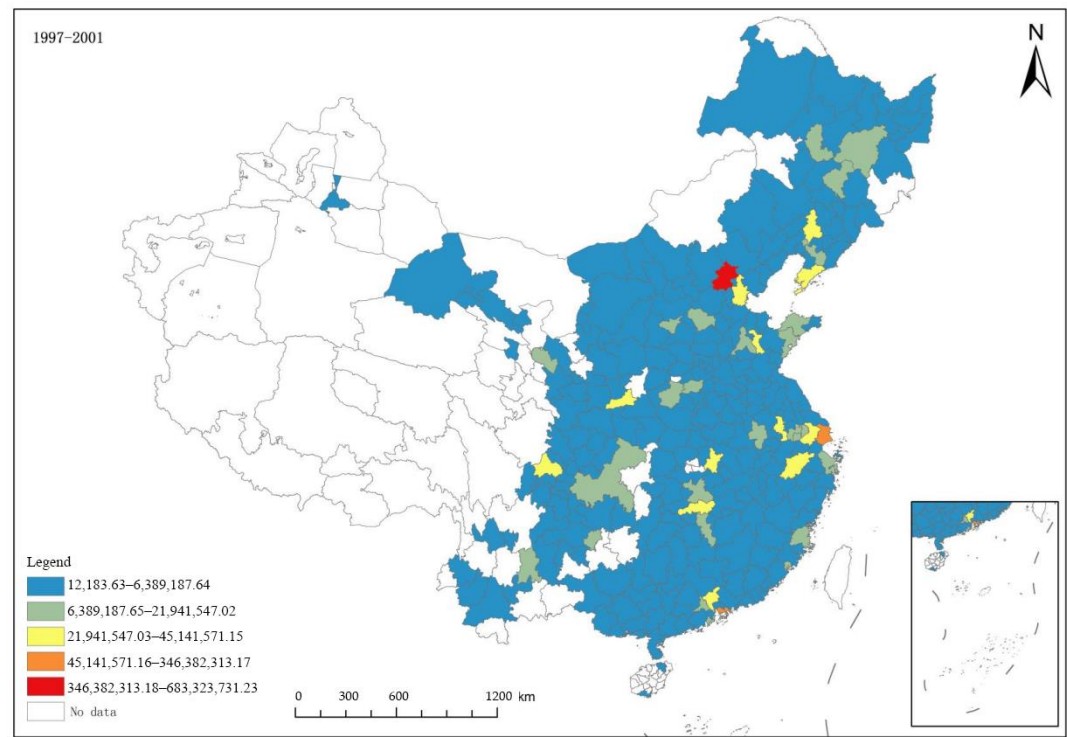

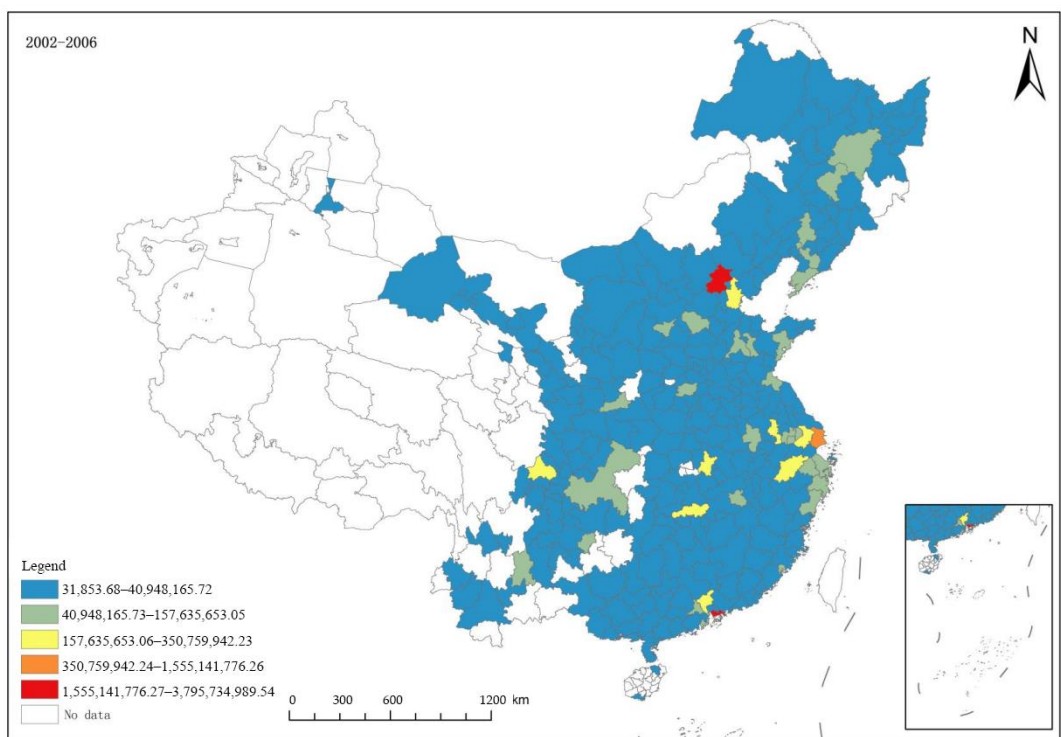

**Figure 4.** *Cont.*

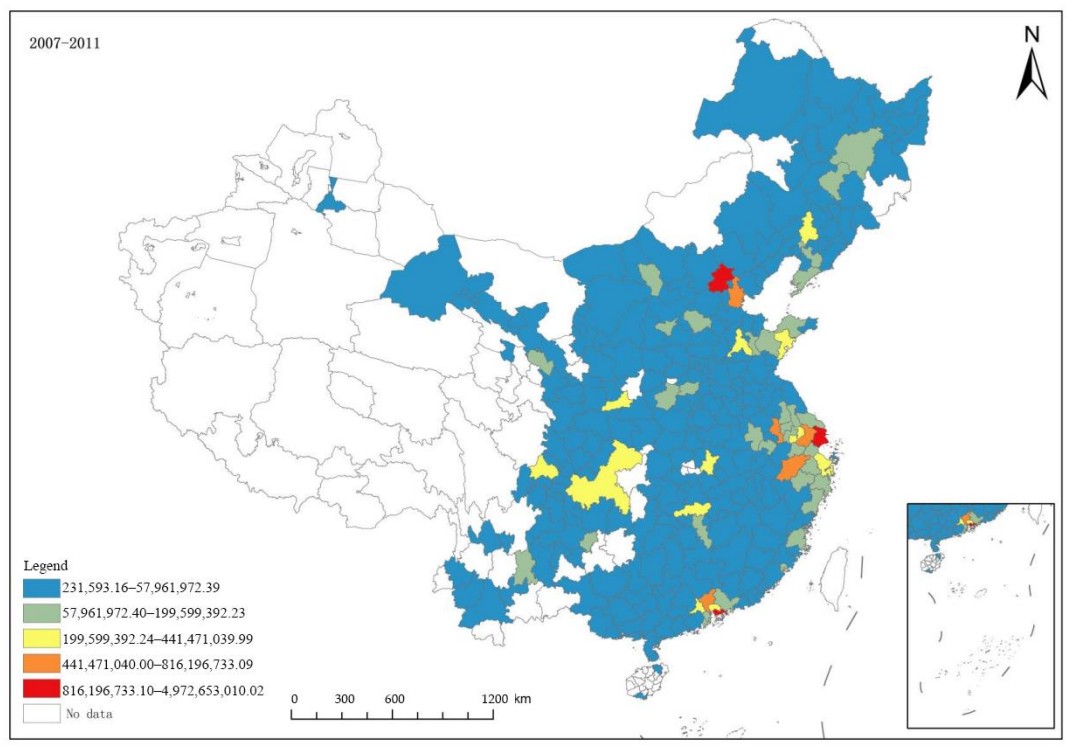

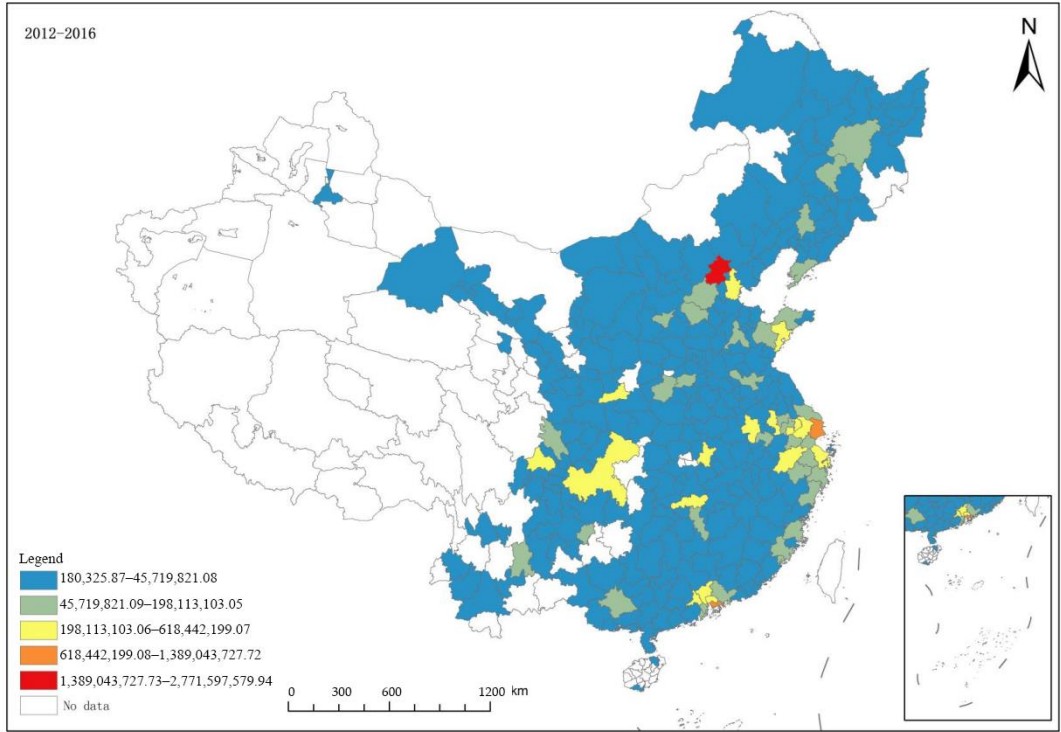

**Figure 4.** *Cont.*

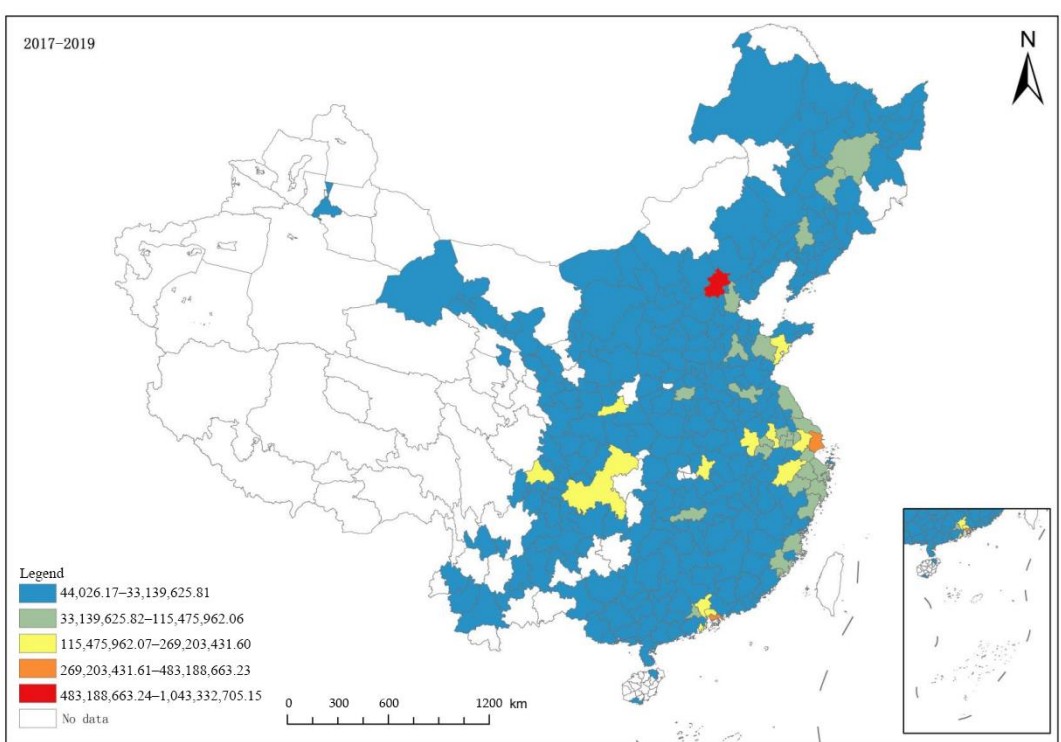

**Figure 4.** Spatial distribution of city-level innovation values in China.

In Figure 4, the regions colored in blue, green, yellow, orange, and red represent cities whose 5-year average innovation values fall within the first category (smallest average innovation values) to the fifth category (largest average innovation values). The figure shows that the innovation values are generally unevenly distributed, and the cities with higher innovation values are mostly concentrated in the Beijing–Tianjin–Hebei area (mainly Beijing and Tianjin), Yangtze River Delta (mainly Shanghai and some surrounding cities), Pearl River Delta (mainly Shenzhen, Guangzhou, and some surrounding cities), and Chengdu–Chongqin area. The degree to which the innovation values are concentrated in these regions is considerably high in that there are only a few cities whose innovation values fall into the first or second category. This indicates the uneven spatial distribution of regional innovation values in China during the sample period.

By comparing each subgraphic in the figure, some temporal characteristics of the distribution of regional innovation values can be inferred. Initially, the degree to which regional innovation values are unevenly distributed is relatively small in the early years, as more regions that fall into the second, third, and fourth categories (colored in orange, yellow, and green in the top left subgraphic, respectively) are observed for the period 1987–1991. However, from 1997 and 2006, the spatial distribution of regional innovation values becomes increasingly uneven in that the innovation values of most cities in the fifth category fall, while only Beijing, Shanghai, and Shenzhen are in the first or second category. In more recent years, the distribution tends to be slightly more even, as the innovation values of some other cities shift into higher categories. In addition, the regions where innovation values are concentrated shift from Northeast to Southeast China. In the earlier years, cities with higher innovation values are found in Northeast China. However, their innovation values gradually decline to the lower categories while the innovation values of cities in the southwest become more notable in the later years.

Overall, the innovation values are unevenly distributed in China during the sample period. Some concentrations of cities with higher innovation values can be observed, while most other cities have relatively lower innovation values, showing that most cities cannot sustain the continuous enhancements of the quality of innovation. The spatial distribution

of the innovation values is not always uneven. In fact, there are multiple cities with higher innovation values, and some of them are concentrated in the northeast in the earlier sample period. The spatial distribution of innovation values becomes increasingly uneven as the concentrated regions shift to the southwest in the middle of the sample period. However, the values tend to be slightly more evenly distributed in more recent years, which provides some evidence for the balanced development of regional innovation in these years.

4.3.2. Spatial Correlation of Regional Innovation Values

In addition to the analysis of the spatial distribution of innovation values, this study analyzes the spatial correlation of regional innovation values based on the local Moran's index as well as the spatial agglomeration effects. The local Moran's index for each city is measured as follows:

$$I_i = \frac{y_i - \overline{y}}{\frac{1}{n}\sum (y_i - \overline{y})^2} \sum_{j \neq i}^{n} w_{ij}(y_i - \overline{y}),$$
(23)

where $I_i$ is the local Moran's index, $n$ is the total number of cities, $w_{ij}$ is the spatial weight between city $i$ and city $j$ (calculated based on the inverse of the geometrical distance between cities), $y_i$ is the innovation value of city $i$, and $\overline{y}$ is the average of all cities' innovation values.

(1) Analysis of the spatial correlation of city-level innovation values

Based on the local Moran's index for each sample city, the sample cities are classified into four groups in terms of the spatial correlation of their innovation values, which also indicates the spatial agglomeration effects of the innovation values, with high–high agglomeration areas (HH cluster), which are areas centered in a city with a high innovation value whose neighboring cities have high innovation values as well; high–low agglomeration areas (HL outlier), in which the center city has a high innovation value while its neighboring cities have low innovation values; low–low agglomeration areas (LL cluster), which are areas in which the central and neighboring cities all have low innovation values; and low–high agglomeration areas (LH outlier), in which the center city has a low innovation value while the neighboring cities have high innovation values.

The local Moran's index for each city's innovation value was calculated using ArcGIS 10.8. Then, based on the results, LISA cluster maps of city-level innovation value in certain sample years were created (Figure 5). In Figure 5, areas colored in pink, red, blue, and light blue represent the HH cluster, HL outlier, LH outlier, and LL cluster, respectively.

As shown in the figure, the HH cluster areas tend to expand during the sample period, and a large HH cluster in the eastern coastal areas centered on the Yangtze River Delta has formed in more recent years. Some other areas, including Tianjin and a few other cities in Guangdong Province in the southern coastal areas, also form an HH cluster after 2011. In the early years, the only HH cluster occurs in Tianjin. The distribution of HH clusters supports the finding that most cities with high innovation values are concentrated in Eastern China, specifically in the Yangtze River Delta and Pearl River Delta, as in the previous discussion.

In contrast, the LL cluster areas have been mostly distributed in Western and Northeastern China in more recent years. Earlier in the sample period, however, few LL clusters can be observed. This further confirms the fact that innovation value in the majority of regions, especially in Western China, is at a low level, which could continue to be the case, since the cities in the vast areas of the LL clusters might negatively influence each other's innovation value.

HL outliers are not common in the earlier years but can be observed in Beijing, the northeastern area, the Chengdu–Chongqin area, and some other western areas in more recent years. Furthermore, the HL outliers are mostly provincial capitals that resources for innovation often flow to from the surrounding cities. Thus, instead of spilling over technologies, knowledge, human capital, and other innovation factors into their neighboring areas, these cities attract innovation factors from their surroundings, resulting in a high level of innovation in these cities and a low level of innovation in their neighboring cities.

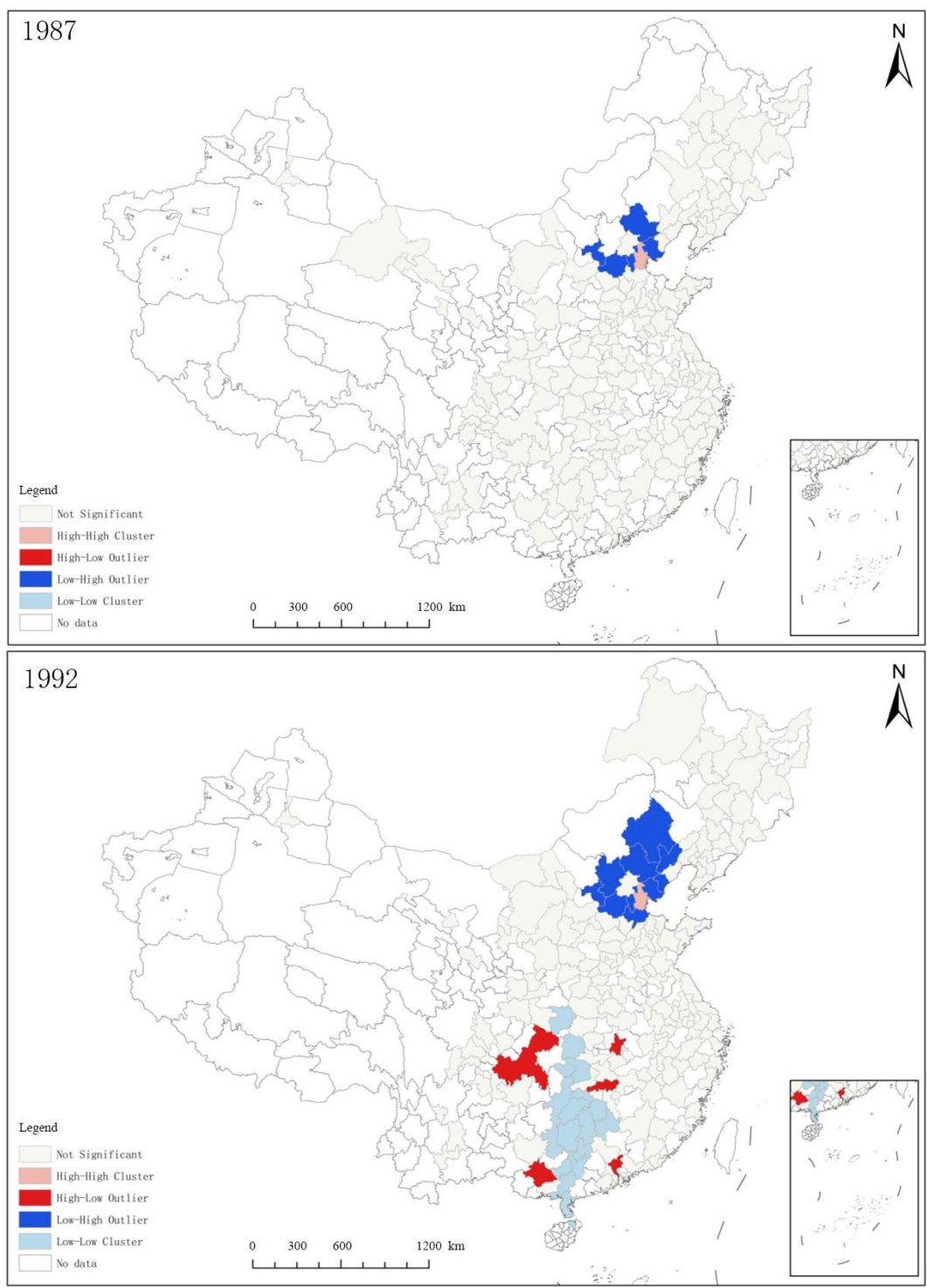

**Figure 5.** *Cont.*

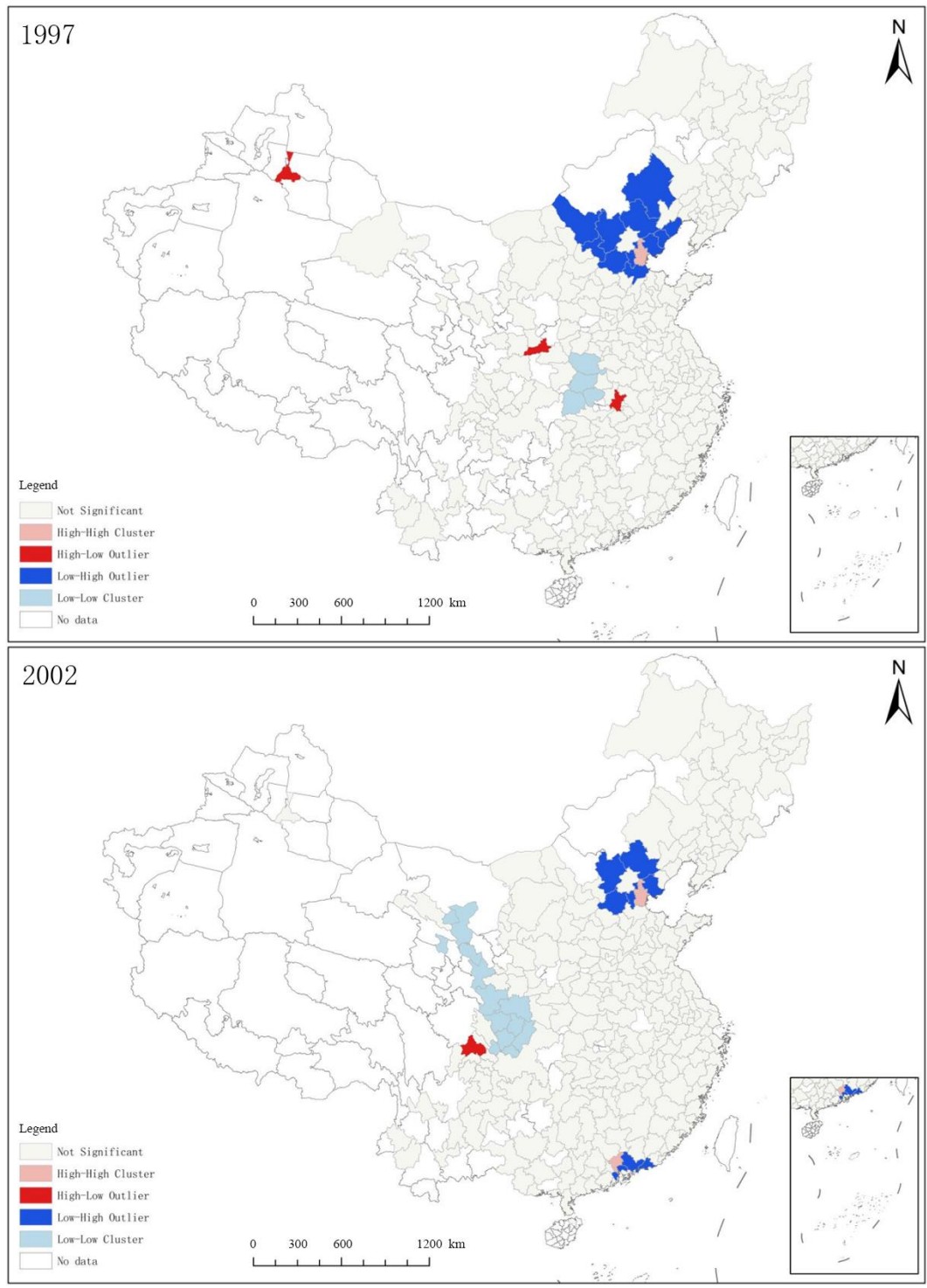

**Figure 5.** *Cont.*

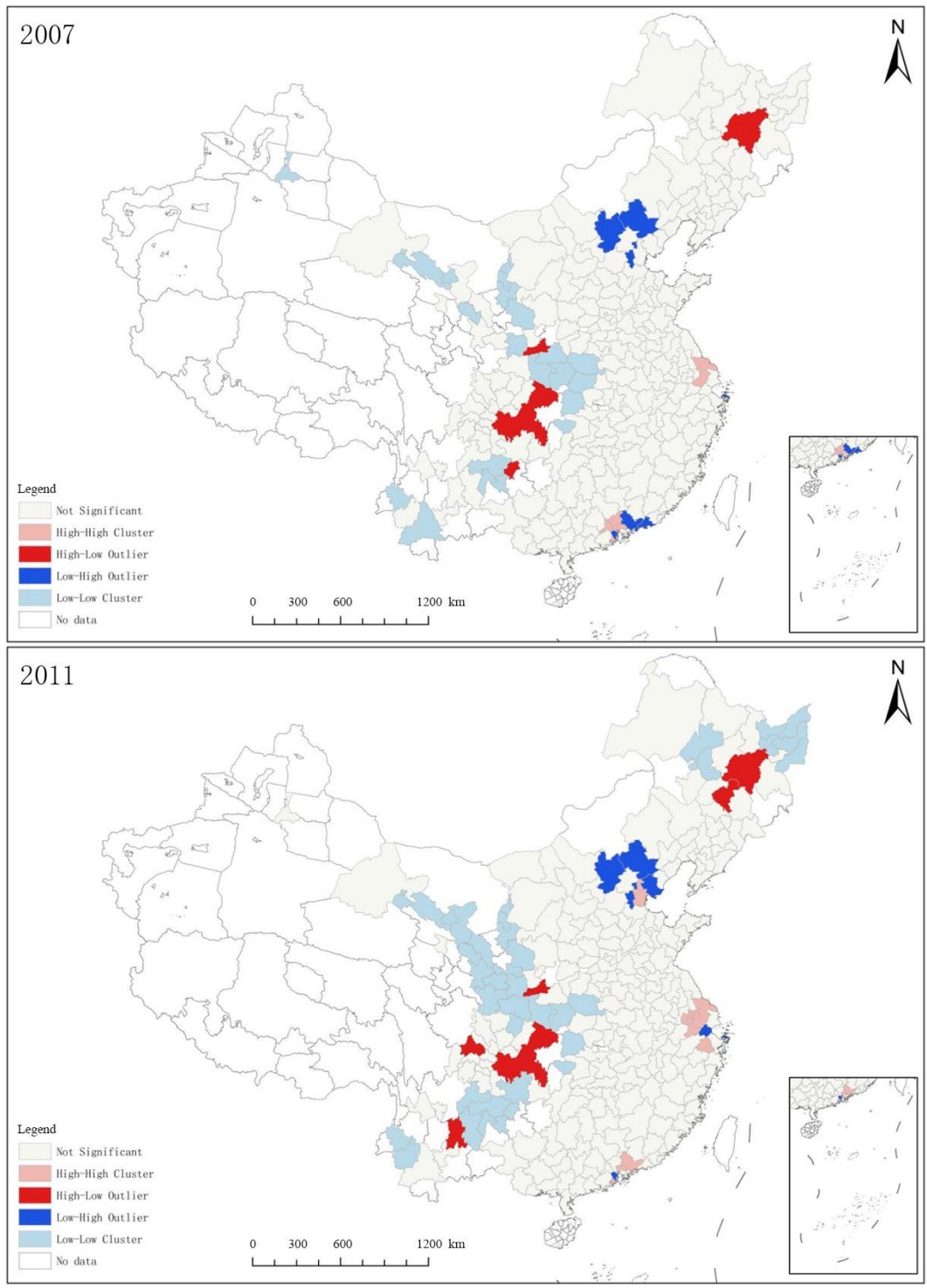

**Figure 5.** *Cont.*

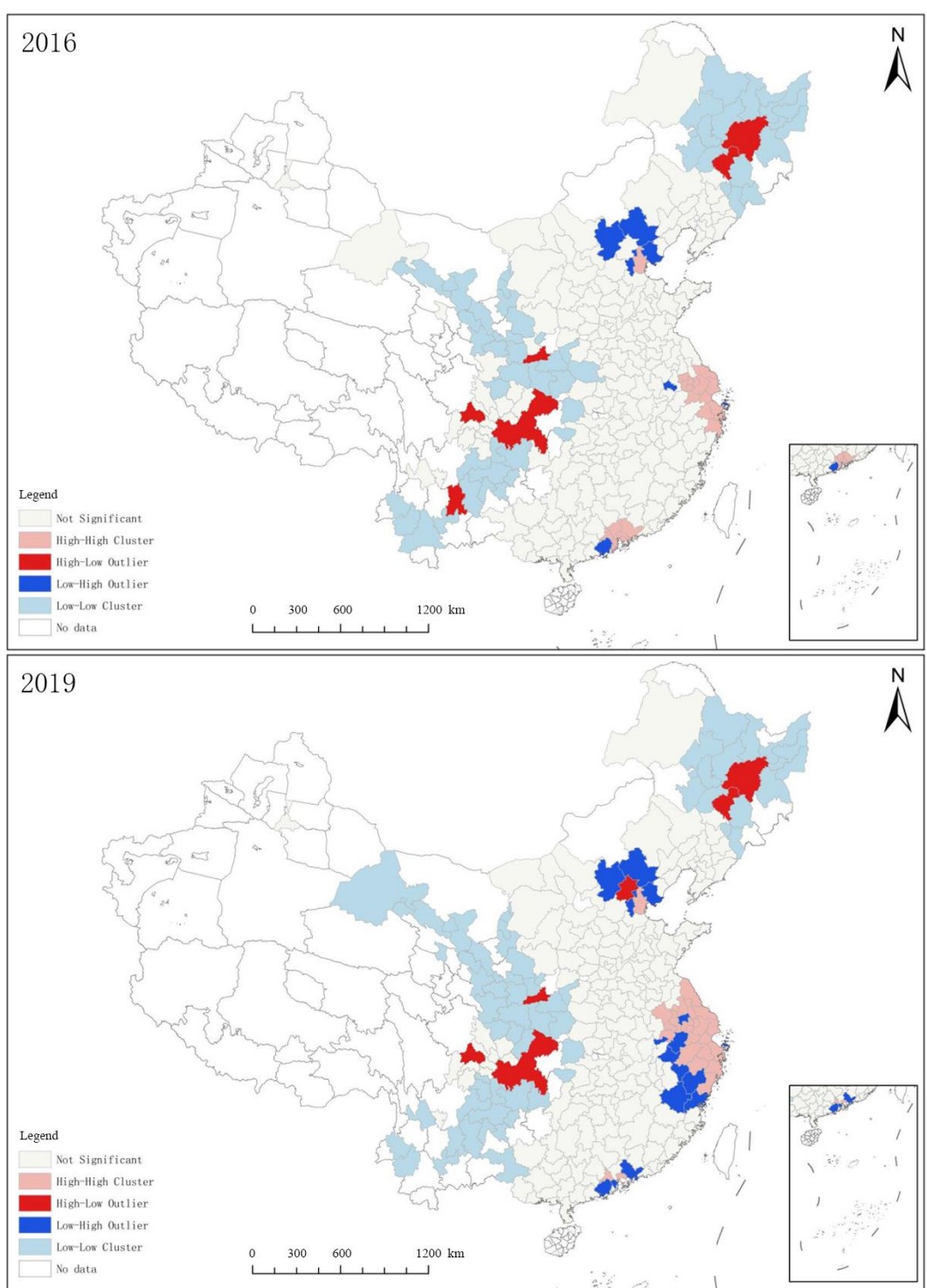

**Figure 5.** LISA cluster maps of city-level innovation values.

The LH outliers are mostly concentrated around Beijing throughout the sample period, while some others lie outside the HH clusters in the Yangtze River Delta in 2019 and the Pearl River Delta after 2002. The cities in the LH outlier areas often have lower innovation value, and their innovation resources tend to flow out into neighboring cities with higher innovation values, possibly seeking higher rent in return. The longer this trend continues, the fewer opportunities these cities will have, leading to a risk of hollowing out. Therefore, action should be taken to increase the innovation ability of cities with low innovation values.

In general, the findings based on the LISA cluster maps of regional innovation values confirm the polarization of innovation values throughout the sample period, which supports the previous findings regarding the uneven spatial distribution of innovation values in China. The polarization of regional innovation values might be attributable to the large LH outlier areas in the early years and the large LL cluster areas in more recent years. The expansion of HH cluster areas in recent years might help reduce the imbalanced overall distribution of innovation values if the synergy of interregional economic development in their surrounding regions can be improved [7]. However, the cumulative innovation capability, including the scale effects in regions with high innovation values, may also widen the gap of overall regional innovation values [7]. In addition, the existence of HL and LH outliers might exacerbate the polarization of innovation values since the high innovation value regions in both HL and LH outliers have countereffects on the level of technological innovation capability in the neighboring regions with low innovation values [8], further lowering the innovation values and causing the unsustainable development of technological innovation if no measures are taken.

(2) Analysis of the spatial correlation of province-level innovation values

To investigate the province-level spatial correlation of innovation values, Moran's index of province-level innovation values is calculated and the corresponding groups based on the spatial correlation patterns are identified. Table 7 presents the resulting classification of provinces into four groups during the sample period.

**Table 7.** Agglomeration of province-level innovation values.

| Year | HH Cluster (Promotion Area) | LH Outlier (Transition Area) | LL Cluster (Low-Level Area) | HL Outlier (Radiation Area) |
|---|---|---|---|---|
| 1987 | Beijing, Tianjin, Hebei, Shanghai Jiangsu, Zhejiang | Jilin, Hainan | Shanxi, Inner Mongolia, Heilongjiang, Anhui, Fujian, Jiangxi, Guangxi, Chongqing, Guizhou, Yunnan, Gansu, Qinghai, Ningxia, Xinjiang | Liaoning, Shandong, Henan, Hubei, Hunan, Guangdong, Sichuan, Shaanxi |
| 1992 | Tianjin, Shanghai, Jiangsu, Zhejiang | Hebei, Jilin, Anhui, Hainan | Shanxi, Inner Mongolia, Heilongjiang, Fujian, Jiangxi, Guangxi, Chongqing, Guizhou, Yunnan, Shaanxi, Gansu, Qinghai, Ningxia, Xinjiang | Beijing, Liaoning, Shandong, Henan, Hubei, Hunan, Guangdong, Sichuan |
| 1997 | Shanghai, Jiangsu | Tianjin, Hebei, Fujian, Hainan | Shanxi, Inner Mongolia, Jilin, Heilongjiang, Anhui, Jiangxi, Henan, Guangxi, Guangxi, Chongqing, Guizhou, Yunnan, Shaanxi, Gansu, Qinghai, Ningxia, Xinjiang | Beijing, Liaoning, Zhejiang, Shandong, Hubei, Hunan, Guangdong, Sichuan |
| 2002 | Shanghai, Jiangsu, Zhejiang, Hunan | Tianjin, Hebei, Fujian, Jiangxi, Guangxi, Hainan | Shanxi, Inner Mongolia, Liaoning, Jilin, Heilongjiang, Anhui, Henan, Hubei, Chongqing, Sichuan, Guizhou, Yunnan, Shaanxi, Gansu, Qinghai, Ningxia, Xinjiang | Beijing, Guangdong, Shandong |
| 2007 | Shanghai, Jiangsu, Zhejiang | Tianjin, Hebei, Anhui, Fujian, Jiangxi, Hunan, Guangxi | Shanxi, Inner Mongolia, Liaoning, Jilin, Heilongjiang, Henan, Hubei, Hainan, Chongqing, Sichuan, Guizhou, Yunnan, Shaanxi, Gansu, Qinghai, Ningxia, Xinjiang | Beijing, Shandong, Guangdong |
| 2012 | Shanghai, Jiangsu, Zhejiang, Anhui, Shandong | Tianjin, Hebei, Fujian, Jiangxi, Hunan, Guangxi, Hainan | Shanxi, Inner Mongolia, Liaoning, Jilin, Heilongjiang, Henan, Hubei, Chongqing Guizhou, Yunnan, Shaanxi Gansu, Qinghai, Ningxia, Xinjiang | Beijing, Guangdong, Sichuan |
| 2017 | Shanghai Jiangsu, Zhejiang, Anhui, Shandong | Tianjin, Hebei, Fujian, Jiangxi, Hunan, Guangxi, Hainan | Shanxi, Inner Mongolia, Liaoning, Jilin, Heilongjiang, Henan, Chongqing, Guizhou, Yunnan, Shaanxi, Gansu, Qinghai, Ningxia, Xinjiang | Beijing, Hubei, Guangdong, Sichuan |
| 2019 | Shanghai, Jiangsu, Jiangsu, Zhejiang, Anhui, Shandong | Tianjin, Fujian, Jiangxi, Hunan, Guangxi, Hainan | Hebei, Shanxi, Inner Mongolia, Liaoning, Jilin, Heilongjiang, Henan, Chongqing, Guizhou, Yunnan, Shaanxi, Gansu, Qinghai, Ningxia, Xinjiang | Beijing, Hubei, Guangdong, Sichuan |

In general, the province-level innovation values in China exhibit an agglomeration effect, although this is primarily the agglomeration of provinces with low innovation values (LL outliers). In 2019, there are 5 provinces in the HH cluster and 15 in the LL cluster, together accounting for 66.67% of the total sample provinces. Most provinces (especially those in Western China) are in the LL cluster, which indicates that the value of provincial innovation is mainly clustered among provinces with low innovation values. The HH clusters occur mostly in provinces in Eastern China (e.g., Shanghai, Jiangsu, Zhejiang). The provinces classified as HL and LH outliers are essentially consistent with the results of the previous analysis of city-level innovation values. Changes in the clustering pattern groups can also be observed in Table 7. Such changes occur mostly between neighboring groups (i.e., neighboring columns in Table 7). For instance, Shandong Province changes from an HL outlier in 2007 to the HH cluster in 2012, and Liaoning and Henan change from HL outliers in 1997 to the LL cluster in 2002. Provinces such as Qinghai, Gansu, Xinjiang, and Ningxia in Western China remain in the LL cluster throughout the sample period.

In summary, there is an agglomeration effect of province-level innovation values in China. However, such agglomeration is mainly characterized by the clustering of provinces with low innovation values, and only a small number of provinces exhibit the agglomeration of high innovation values. On one hand, the provinces in the HH clusters that are mutually promoting the sustainable improvement of technological innovation are in the Yangtze River Delta, which is consistent with previous findings. On the other hand, the large number of provinces in the LL clusters indicates that many regions in China are unable to promote sustainable growth in their innovation values. In addition, the changes in the groups of clustering patterns are not significant, and most provinces remain in the same clustering groups throughout the sample period. Therefore, regarding the spatial correlation analysis of provincial innovation values in China, it can be concluded that only a few provinces achieve innovation sustainability, while the majority of provinces are unable to develop technological innovation sustainably and remain at low innovation values.

(3) Further analysis of the spatial agglomeration of regional innovation values

Using the hotspot analysis tool in ArcGIS 10.8, the regions are categorized into extreme hot spots, hot spots, sub-hot spots, sub-cold spots, cold spots, and extreme cold spots based on the Getis–Ord Gi* calculation (Getis–Ord Gi* is calculated based on city-level innovation values using ArcGIS 10.8). Figure 6 shows the results, in which areas colored red, orange, and light orange are, respectively, extremely hot, hot, and sub-hot spots, while those in blue, light blue, and light green are extremely cold, cold, and sub-cold spots.

The figure indicates that there are only a small number of hot spots of regional innovation values in China, with a stepped distribution of hot spots according to extreme hot spots, hot spots, and sub-hot spots. In 2019, there are only two hot spots, both located in Eastern China. This is closely related to China's industrial base. For mature industries, technological evolution is close to the saturation line of the technological paradigm, with relatively fewer technological opportunities and fewer innovation achievements. Meanwhile, China's integration into the international division of labor in its previous round of economic development was based on the demographic dividend and comparative cost advantages. This brought about rapid growth in export trade while determining that China's pre-labor-intensive industries were better developed, placing its industrial structure on the lower end and affecting its innovation efficiency.

In terms of temporal evolution, the hotspot agglomerations show a trend of increasing in number and expanding in scope, with the hotspot agglomerations shifting from Northeast to Northern, Eastern, and Southern China. This is because the northeast region previously had a better industrial base but was mostly dominated by high-energy-consuming, high-polluting industries, which had less room for technological innovation as the production methods matured. Eastern and Southern China, meanwhile, have used their location, economic base, and distribution of educational resources to attract innovative talent and enterprises to cluster in the region, while optimizing their industrial structure and releasing innovation space, thereby expanding the scope of hotspot clusters.

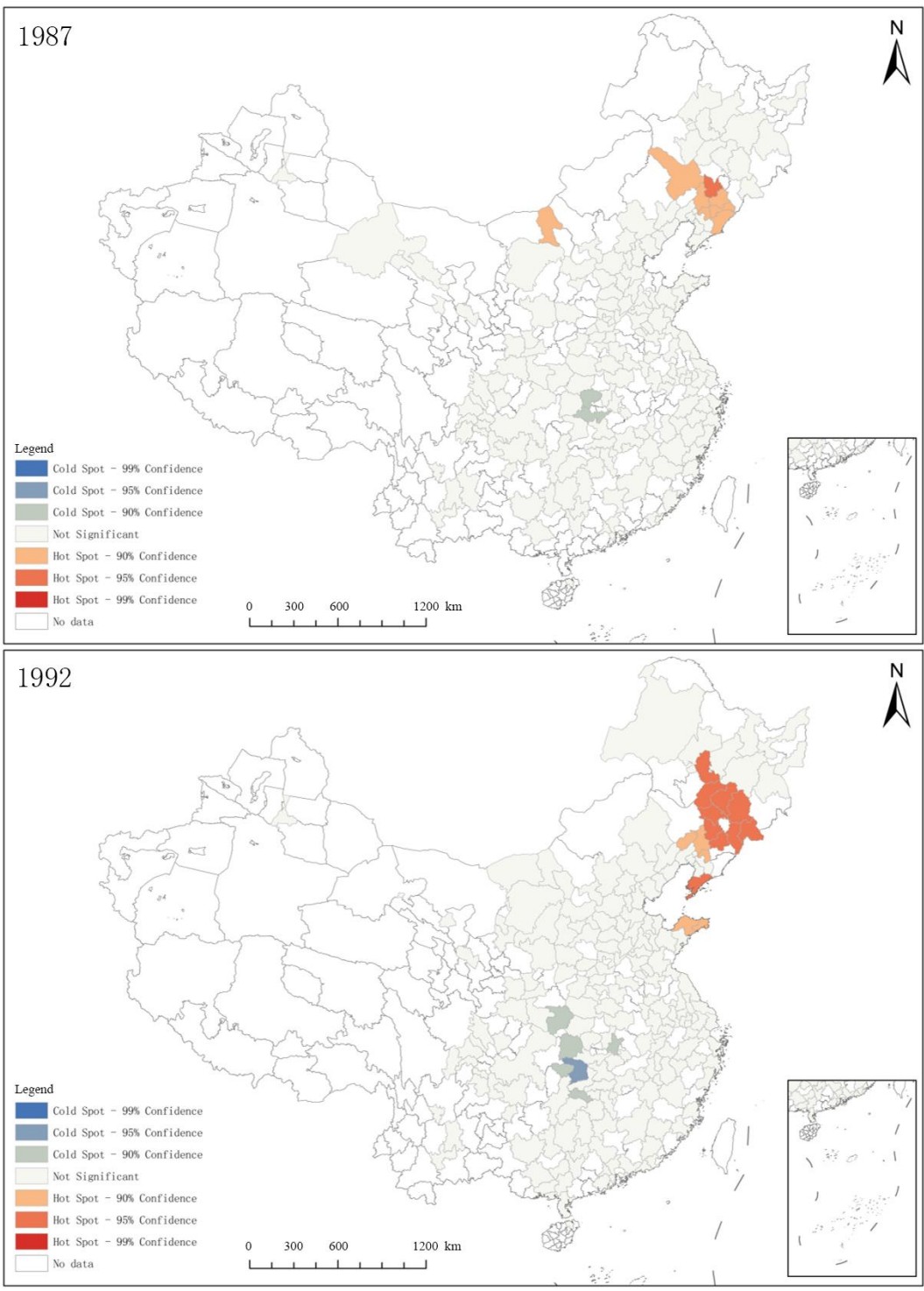

**Figure 6.** *Cont.*

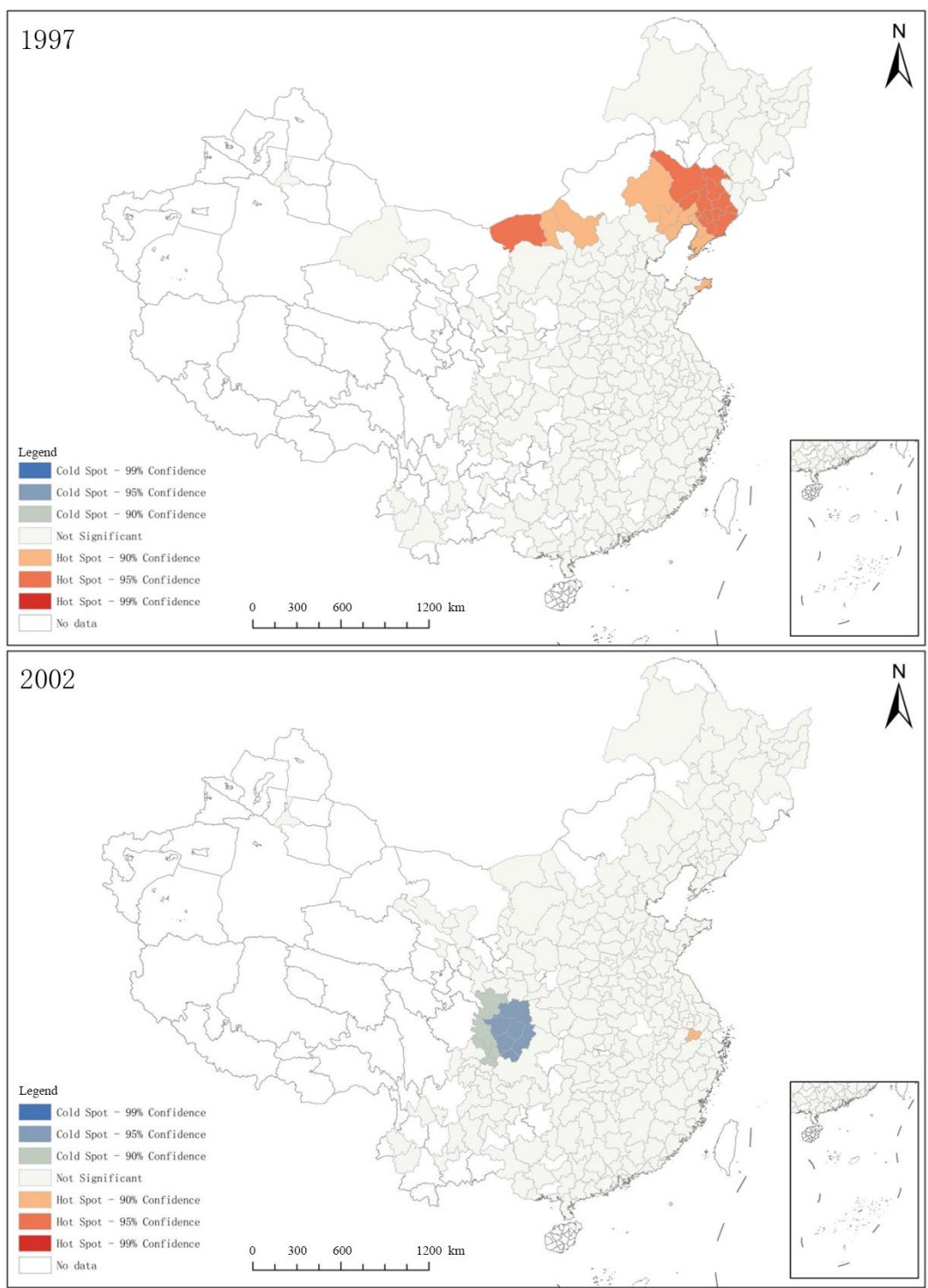

**Figure 6.** *Cont.*

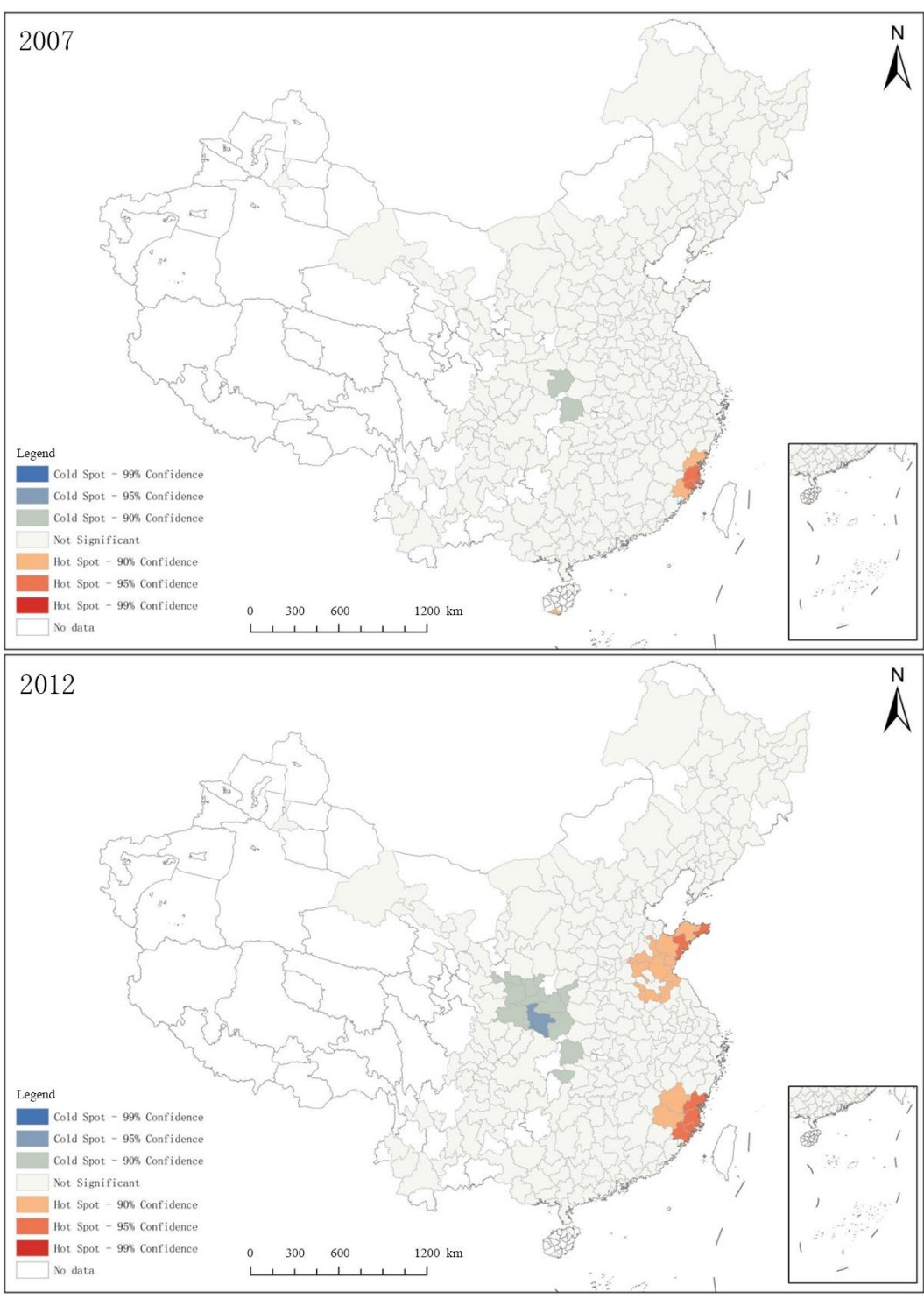

**Figure 6.** *Cont.*

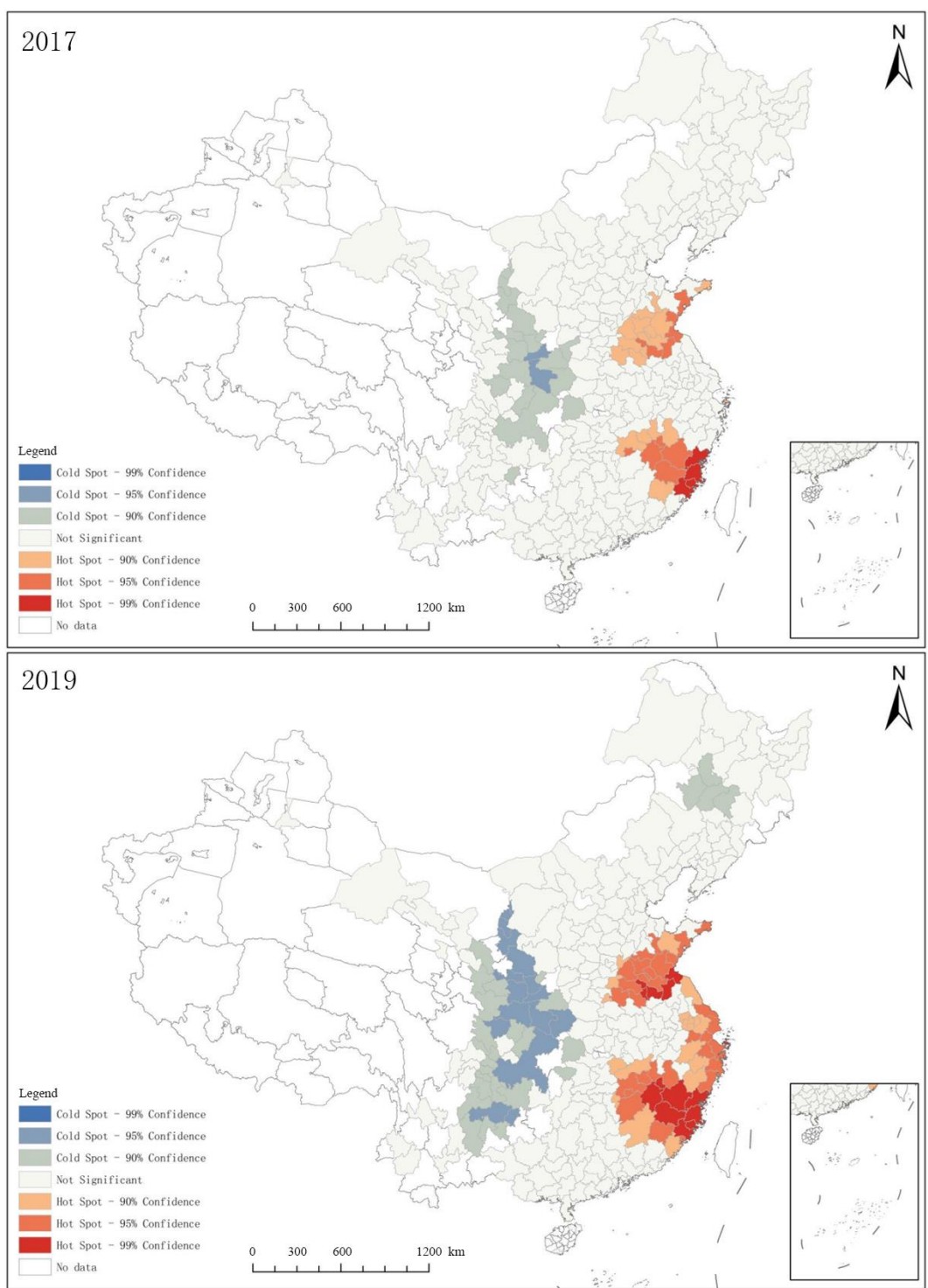

**Figure 6.** Hot-spot maps of regional innovation values.

In accordance with the analyses based on the LISA cluster map of city-level innovation values and the spatial correlation of province-level innovation values in the previous section, the hot-spot analysis further asserts that only a small number of regions (mainly eastern coastal regions) in China in the sample years achieve relatively high innovation values and exhibit the clustering of high innovation values, and most other regions' innovation values are low and exhibit the agglomeration of low innovation values. Therefore, the second hypothesis (H2) is confirmed. Regions with geographic concentrations of innovative activity often benefit from economies of scale related to the innovative inputs, market size,

and knowledge spillover, which are conducive to long-term endogenous growth in such regions [88]. Therefore, it is expected that regions of agglomeration of high innovation values can promote sustainable technological innovation. On the contrary, regions with low innovation value clusters may struggle in improving their innovation values, meaning they cannot sustain their technological advancement due to the lack of adequate innovation inputs, sizeable markets, and accumulation of knowledge. Although the Chinese government has implemented a series of policies promoting regional technological innovation, such as establishing national high-tech industrial parks and issuing innovation city pilot policies, such policies may have relatively minor effects on regional innovation efficiency [5,6], may influence limited geographical regions [4], or may exert more significant impacts on relatively developed cities such as the capital city of a province [6]. Therefore, regions of high innovation value agglomeration, which are endowed with more resources of innovation inputs and have better innovation and economic environments, may take advantage of such policies and further sustain the technological innovation, while the influence of such policies has not yet emerged in regions of low innovation value clusters.

## 5. Conclusions and Policy Recommendations

Using an improved regional innovation value model, this study estimated the regional innovation values in China using invention patents. Then, the study explored the spatial distribution of regional innovation values during the sample period. The findings are summarized below.

First, the proposed regional innovation value model provides an effective way to estimate parameters related to the distribution of invention patent values while incorporating the effects of regional innovation factors. Thus, the model can be used as an effective way to estimate regional innovation values, providing insights for regional innovation and sustainability. Second, nationwide, the innovation value and average value of invention patents in China show a temporal trend of first increasing and then decreasing. In addition, only a small number of cities achieve sustainability in the development of technological innovation, since the innovation levels of a few cities with the highest innovation values are much greater than the other cities. Moreover, Eastern China has a much higher innovation value than Central, Western, and Northeastern China, and such differences expanded during the earlier years and started to narrow after 2008. Third, the evidence shows that the high innovation values have mostly clustered in eastern coastal regions, while other regions have mainly experienced an agglomeration of low innovation values, which indicates that only a few regions have achieved sustainable technological innovation in terms of their innovation value. In addition, some regions have experienced a hollowing out of innovation resources since their neighboring regions have much higher innovation values, which has resulted in unbalanced and unsustainable development of technological innovation across the regions. Fourth, a few regions in Eastern China mutually promote sustainable innovation growth, as cities in these regions have highly concentrated innovation values. However, similar to the above findings, most other regions do not have concentrated innovation values or even have very low levels of concentrated innovation values, as in Western China.

Some policy implications can be drawn based on the findings. First, given China's lack of high-quality innovation and its low level of innovation value clustering, China needs to continue to promote its innovation-driven development strategy and focus on cultivating high-value innovation to sustain the continuous development of technological innovation. Second, to address the regional innovation value gap caused by resource endowments, innovation environmental factors, and other economic factors, there should be an emphasis on interregional industrial transfer; improving infrastructure access in underdeveloped regions; strengthening the attractiveness of the central, western, and northeastern regions for talent; and improving the innovation capacity of underdeveloped regions. Lastly, it is necessary to overcome regional monopolies, give full play to spatial spillover effects, and strengthen the interregional exchange of innovation information in order to achieve a

balanced and sustainable development path for regional innovation, followed by balanced regional economic growth in the long run.

## 6. Limitations and Future Researches

There are a few aspects of this paper that can be extended to future studies. First, due to data limitations, other types of innovation outputs such as new products and services, the improvement of production processes, and other inventions and scientific discoveries that are not patented were not addressed in this paper. Future researches may investigate and evaluate different types of innovation in order to provide estimations that can better reflect the value of regional innovation. Second, examining the causality between innovation inputs (along with the factors of innovation and the economic environment) and innovation output values can help the innovators and policy makers to identify the interrelationships between resources and innovation outcomes, allowing them to efficiently allocate resources and provide sufficient incentives for innovative activities. Future research studies can focus on analyzing the causality. Third, the Chinese government has implemented a series of policies, including the establishment of high-tech industrial parks and national innovative cities, to stimulate innovation. Although scholars have investigated their effects on the innovation efficiency and capacity in regions across China, few studies have investigated such policies' potential effects on the value of regional innovation, which would be of interest for future research studies.

**Funding:** This research received no external funding.

**Institutional Review Board Statement:** Not applicable.

**Informed Consent Statement:** Not applicable.

**Data Availability Statement:** All data utilized in this study are from public resources and have been cited with references accordingly.

**Conflicts of Interest:** The author declares no conflict of interest.

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
