# Peer review of "The Sustainability of Regional Innovation in China: Insights from Regional Innovation Values and Their Spatial Distribution"

_sustainability, doi:10.3390/su151310398_

Round 1
Reviewer 1 Report
The literature review needs to be presented.
Currently, introduction is only addressing review of literature. It needs to present organized literature reivew.
Author also needs to present research hypotheses.
Also, the variables need to be theory based.
Author Response
Dear Reviewer,
Thank you very much for carefully reading the manuscript. I am very grateful for your helpful and valuable recommendations to further improve the paper. The manuscript has been carefully revised based on your comments and suggestions. All the revised contents have been marked in red in the manuscript, and the responses to your comments have been marked in red in this response letter. More details are as follows.
- The literature review needs to be presented.
Response: Thank you very much for your suggestion. The sections in this paper have been reorganized and the section of Literature Review and Research Hypothesis is rewritten and presents following the Introduction section. In addition, a few literatures related to the research top have been supplemented to improve the review of existing studies.
- Currently, introduction is only addressing review of literature. It needs to present organized literature review.
Response: Thank you very much for your valuable comments. The Introduction section has been revised to properly introduce this paper in general. Discussion related to literature reviews that was originally presented in the Introduction section has now been removed. And the section of Literature Review and Research Hypothesis is organized based on the literatures related to the estimation of innovation value and the analysis of the spatial distribution of regional innovation.
- Author also needs to present research hypotheses.
Response: Thank you very much for pointing out this issue. I sincerely apologize for missing the research hypotheses in this paper. Two major research hypotheses are now added in section 2., which are the followings: Hypothesis 1, the regional innovation value model that incorporates the patent value and a regional innovation indicator system can effectively estimate the regional innovation values in China; and Hypothesis 2, the spatial distribution of regional innovation value is unbalanced and mainly characterized by the agglomeration of regions with low innovation values. In addition, the responses to the two hypotheses are presented in the section of Results and Discussion.
- Also, the variables need to be theory based.
Response: Thank you very much for the valuable advice. A review of the theory of regional innovation system has been added in the section of Literature Review in order to establish the theoretical foundations of variable selection. In addition, a detailed discussion on the theoretical relationships between the indicators of innovation inputs, innovation environment, and economic environment (as well as the associated secondary (tier 2) indicators) and the regional innovation values (estimated based on patent values) has been added prior to Table 1 (which presents the regional innovation indicator system conducted in this paper).
Reviewer 2 Report
This study estimates regional innovation values based on an enhanced regional innovation value model that incorporates patent values and a system of regional innovation indicators. Data for invention patents as well as regional innovation indicators in 282 cities from 1987 to 2019 in China are used for estimation. The article is well written and easy to understand, the topic is current and China is one of the countries that are emerging in the area of innovation.
Author Response
Dear Reviewer,
Thank you very much for carefully reading the manuscript. I am very grateful for comments on the presentation and topic selection in this paper. In addition, the manuscript has been further revised to improve the presentation of the research. Thanks again for your review of this paper.
Reviewer 3 Report
Dear Author,
Thank you for this interesting and inspiring article. After carefully reviewing the manuscript, I conclude that your study is highly relevant to sustainability research and therefore well suited for publication in this journal. Your findings contribute to research that addresses the measurement of innovation performance in a regional context and provide both new theoretical and empirical arguments that I greatly appreciate. The methodological work, both in terms of theoretical modeling and empirical estimation, is very solid. Nevertheless, I have some suggestions for improvement, as some points seem to me unclear or not sufficiently addressed.
First of all, I would like to note that the definition of the term "value" seems somewhat unclear to me. What exactly is meant by value? If I understand you correctly, innovation value primarily refers to the value of the patent. This should be clearly communicated. You should also adjust the language in the appropriate places, for example in the research question. As currently formulated, your research question seems somewhat circular to me: "How can the value of regional innovation be accurately measured in terms of its value?"
I also find it somewhat irritating that you talk about regional innovation value, but only use patents for the calculation. Innovation value is conceptually broader than patent value. This has also been discussed extensively in the relevant literature for a long time. In my view, you either have to provide very good arguments why you assume that patent value and regioanl innovation value are strongly correlated, or you have to go into more detail on this limiting aspect in the limitation section. By the way, I do not believe that patents are a good indicator to fully reflect regional innovation efforts.
Furthermore, I would appreciate it if you could address the causality direction between innovation inputs and patents in the discussion. The line of causality does not necessarily have to be unidirectional, which would cause methodological problems for the estimation.
Despite these comments, however, I would like to conclude by saying that I already liked the article overall and see great potential. I would therefore like to encourage you to reconsider the points I mentioned and, if applicable, implement them in a revised version of the article.
Thank you very much.
Author Response
Dear Reviewer,
Thank you very much for carefully reading the manuscript. I am very grateful for your helpful and valuable recommendations to further improve the paper. The manuscript has been carefully revised based on your comments and suggestions. All the revised contents have been marked in red in the manuscript, and the responses to your comments have been marked in red in this response letter. More details are as follows.
First of all, I would like to note that the definition of the term "value" seems somewhat unclear to me. What exactly is meant by value? If I understand you correctly, innovation value primarily refers to the value of the patent. This should be clearly communicated. You should also adjust the language in the appropriate places, for example in the research question. As currently formulated, your research question seems somewhat circular to me: "How can the value of regional innovation be accurately measured in terms of its value?"
Response: Thank you very much for your suggestion. Based on your suggestions, the term “value” in the context of innovation is clarified in the beginning of Literature Review and Research Hypothesis section. In addition, the reason why the value of innovation is limited to patent value is provided in the same section. Also, the presentation of the research question that is improperly phrased (in the second to last paragraph in the Introduction section) has been adjusted accordingly.
I also find it somewhat irritating that you talk about regional innovation value, but only use patents for the calculation. Innovation value is conceptually broader than patent value. This has also been discussed extensively in the relevant literature for a long time. In my view, you either have to provide very good arguments why you assume that patent value and regional innovation value are strongly correlated, or you have to go into more detail on this limiting aspect in the limitation section. By the way, I do not believe that patents are a good indicator to fully reflect regional innovation efforts.
Response: Thank you very much for pointing out this issue. The main reason why only patent information was utilized to estimate regional innovation value is the limited accessibility of data for other types of innovation outputs, such as new products and services, improved productivities based on process innovation, and even the inventions and scientific discoveries, which has been discussed in the Literature Review section. However, as you have pointed out, patent information as an indicator of innovation cannot fully reflect regional innovation efforts. This is indeed the limitation of this paper, and is addressed in the section of Limitation.
Furthermore, I would appreciate it if you could address the causality direction between innovation inputs and patents in the discussion. The line of causality does not necessarily have to be unidirectional, which would cause methodological problems for the estimation.
Response: Thank you very much for your valuable advice on the direction of causality between innovation inputs and patents. An addition paragraph discussing the potential direction of such causality is added following equation (8), in which the production of innovation outputs is describe. In addition, a few relevant literatures have been supplemented, although their findings about the direction of causality between innovation inputs and outputs are somewhat ambiguous.
Reviewer 4 Report
This manuscript examines the sustainability of regional innovation in China. The research selection is of some value, but the following issues still need to be improved:
1、Line 45 states "which suggests an unsustainable development path of innovation in China." However, this conclusion cannot be drawn from the above literature. Why does the spatial clustering of innovation lead to unsustainable innovation? How does the link between innovation presenting spatial agglomeration, which may be caused by the construction of technology industrial parks or the restructuring of urban planning, such as whether it is sustainable, manifest itself?
2、Lines 125 to 134 are not a description of the methodology, and the main research logic and research steps of the methodology section should be explained in this section. It is recommended that this section be amended.
3、In the methods section the presentation should be reduced for those parts that have not been improved and only use existing proven models, and there is no need to repeat the formulas of existing generic analysis methods. The presentation of model optimisation content should be increased
4、The indicators are given directly in Table 1, but it is not found in the article how the selection of indicators was carried out, nor is the rationality of the selection expressed, At the same time an indicator cannot be sourced from just one piece of literature, as this may lead to unreliable results. And it is recommended that this section be improved.(Secondary indicators should be sourced specifically and not be cited en masse as primary indicators. It is also important to justify the reasonableness of the indicators selected, as there may be some endogeneity issues with the indicators selected in different articles, and some overlap in the nature of what the indicators cover.)
5、It is already 2023, so why is the data for the analysis selected only up to 2019? If the data for 2022 has not yet been fully uploaded, it would be possible to analyse up to 2021.
6、The presentation in the manuscript is recommended to follow the fixed presentation in the research paper.
7、The study concludes that agglomeration of innovation may lead to unsustainable innovation, but that agglomeration of innovation may come to a large extent from policy orientation and support, such as the establishment of innovation industrial parks, innovation cities, etc. Why does agglomeration of innovation lead not to scale effects that drive innovation but to unsustainability? It is suggested that the authors provide an answer to this question.
8、The study concludes that the value of innovation in China shows less clustering of high value innovations, but that the clustering of high innovation may be largely due to policy direction and support, such as the establishment of innovation industrial parks, innovation cities, etc., and is due to the existence of policy support and tax support resulting in a sustained increase in the value of innovation over time. If this is the case, do the existing conclusions hold true, as the value of innovation that should belong to other regions is incorrectly counted in the city where the innovation park is located? It is suggested that the authors provide an explanation for this situation.
Moderate language proofing required
Author Response
Dear Reviewer,
Thank you very much for carefully reading the manuscript. I am very grateful for your helpful and valuable recommendations to further improve the paper. The manuscript has been carefully revised based on your comments and suggestions. All the revised contents have been marked in red in the manuscript, and the responses to your comments have been marked in red in this response letter. More details are as follows.
1、Line 45 states "which suggests an unsustainable development path of innovation in China." However, this conclusion cannot be drawn from the above literature. Why does the spatial clustering of innovation lead to unsustainable innovation? How does the link between innovation presenting spatial agglomeration, which may be caused by the construction of technology industrial parks or the restructuring of urban planning, such as whether it is sustainable, manifest itself?
Response: Thank you very much for your valuable suggestions. The misleading discussion in the Introduction section as you mentioned is now clarified. Some additional literatures related to the clustering of regional innovation has been supplemented in this section. In particular, literatures suggested that there exist agglomeration effects of innovation across regions in China. Yet the overall spatial distribution of regional innovation is disparate and characterized by mainly the clustering of regions with low innovation capability, which indicates that a large number of regions in China have been struggling with low capabilities of technological innovation, and thereby unable to sustain technological advancement. Although the implementation of policies such as the establishment of high-tech industrial parks and national innovative cities exerts positive impact on regional innovation, such effect may be more significant in relatively developed regions. In addition, the findings in this paper support the conclusion of the clustering of low regional innovation values as the values of innovation in most regions in China are low, and their neighboring regions are in the similar condition.
2、Lines 125 to 134 are not a description of the methodology, and the main research logic and research steps of the methodology section should be explained in this section. It is recommended that this section be amended.
Response: Thank you very much for your valuable advice on the discussion of methodology. The main research logic and steps in this paper has been discussed in the beginning of Methods and Data section.
3、In the methods section the presentation should be reduced for those parts that have not been improved and only use existing proven models, and there is no need to repeat the formulas of existing generic analysis methods. The presentation of model optimisation content should be increased
Response: Thank you very much for your valuable comments. The discussion on the original analytic works of patent renewal model has been reduced, while some of the formulas describing the essential logic and steps in estimating patent values are kept. In addition, the discussion on the optimization problems faced by the representative agent, including the description of the setups and the solution, is extended.
4、The indicators are given directly in Table 1, but it is not found in the article how the selection of indicators was carried out, nor is the rationality of the selection expressed, At the same time an indicator cannot be sourced from just one piece of literature, as this may lead to unreliable results. And it is recommended that this section be improved. (Secondary indicators should be sourced specifically and not be cited en masse as primary indicators. It is also important to justify the reasonableness of the indicators selected, as there may be some endogeneity issues with the indicators selected in different articles, and some overlap in the nature of what the indicators cover.)
Response: Thank you very much for your valuable suggestion. A detailed discussion on the selection of indicators has been presented prior to Table 1. In this discussion, the rationality and the theoretical foundations of the selected indicators are presented, while two main sources of literatures (“Evaluation Report on China’s Regional Innovation Capacity 2021” and “Evaluation Report on the Innovation Capacity of National Innovative Cities 2021”, both are published by Scientific and Technical Documentation Press) that conducted similar indicator systems are cited as the guidance of selection. Furthermore, the selection of each secondary (tier 2) indicators are discussed in detail including the rationale of selection and their theoretical relationship with the value of innovation to avoid potential endogeneity issues.
5、It is already 2023, so why is the data for the analysis selected only up to 2019? If the data for 2022 has not yet been fully uploaded, it would be possible to analyse up to 2021.
Response: Thank you very much for your valuable suggestion. Additional information about the sampling time span is now added as a footnote (footnote 21) in the section describing patent data. The data for the invention patents were collected in 2022. However, since the examination procedure of an invention patent typically takes three years after its application in China, those that were applied for after 2019 were mostly under examination, and their legal status were not observable upon the time of data collection. Therefore, the invention patents applied for after 2019 were excluded from the data set.
6、The presentation in the manuscript is recommended to follow the fixed presentation in the research paper.
Response: Thank you very much for pointing out this issue. I sincerely apologize for the poor organization of the presentation in this paper. The section of Literature Review and Research Hypotheses and Limitations and Future Researches are now added. The originally sepeerated sections of empirical results and analyses of the spatial distribution of regional innovation values are now combined as the Methods and Data section. Moreover, the presentation has been reorganized in the order of Introduction, Literature Review and Research Hypotheses, Methods and Data, Results and Discussion, Conclusions and Policy Recommendations, and Limitations and Future Researches.
7、The study concludes that agglomeration of innovation may lead to unsustainable innovation, but that agglomeration of innovation may come to a large extent from policy orientation and support, such as the establishment of innovation industrial parks, innovation cities, etc. Why does agglomeration of innovation lead not to scale effects that drive innovation but to unsustainability? It is suggested that the authors provide an answer to this question.
Response: Thank you very much for your valuable suggestion. Some discussions on the agglomeration of innovation value, along with a few supplementary literatures, are added to the parts that analyze the spatial distribution and agglomeration of regional innovation values (section 4.3). In general, the findings in this paper suggested that the spatial distribution of regional innovation values is mainly characterized by the agglomeration of low innovation values, although a small number of regions exhibit the agglomeration of high innovation values. The governmental support to innovation through policies such as the establishment of high-tech industrial parks and innovative cities does have positive effects on regional innovation. Yet literatures show that these effects may be ambiguous and even limited in some aspects. Therefore, it is possible that regions with low innovation value and are clustered with other similar regions may be not on the path of sustainable technological growth. In addition, lacking an empirical examination on the effect of such policies on the regional innovation value does not provide a concrete conclusion in this paper. Therefore, such limitation is addressed in the last section in this paper and can be addressed in the future.
8、The study concludes that the value of innovation in China shows less clustering of high value innovations, but that the clustering of high innovation may be largely due to policy direction and support, such as the establishment of innovation industrial parks, innovation cities, etc., and is due to the existence of policy support and tax support resulting in a sustained increase in the value of innovation over time. If this is the case, do the existing conclusions hold true, as the value of innovation that should belong to other regions is incorrectly counted in the city where the innovation park is located? It is suggested that the authors provide an explanation for this situation.
Response: Thank you very much for your valuable suggestion. A discussion with some complementary literatures regarding the lacking of high value innovations in regions in China is added at the end of Results and Discussion section. As some studies indicated, policies including the establishment of industrial parks and innovation cities have positive impact on local innovative activities through the subsidies for innovative activities and reduction of taxation, which will result in the increase in regional innovation value. As a matter of fact, based on the estimated innovation values at city and area level (see section 4.2), the overall innovation values in China increased dramatically compared to those in the early sample years, which provides indirect evidence on the positive effects of such policies. Yet the regional innovation value may be affected by multiple factors of the innovation environment and economic condition in a region. Some other exogenous shocks (such as technology blockade), which are not addressed in this paper due to the limited space for discussion, may also restrict China’s technological advancement and hedge the effort of government’s innovation policies. To further investigate the impacts of government’s innovation policies (and possible other exogeneous factors), it is necessary to conduct a systematic empirical analysis addressing these factors, which is a limitation of this paper. This limitation is addressed in the last section. In addition, conducting such analysis may also verify whether the innovation value is correctly counted in regions where the high-tech industrial parks and/or innovation cities are located.
Round 2
Reviewer 4 Report
The manuscript can be accepted. The author has made good changes to the content.